# UP-DOWN cortical dynamics reflect state transitions in a bistable network

Daniel Jercog[1‡*], Alex Roxin[2], Peter Barthó[3], Artur Luczak[4], Albert Compte[1†], Jaime de la Rocha[1†*]

[1]Institut d'Investigacions Biomèdiques August Pi i Sunyer, Barcelona, Spain; [2]Centre de Recerca Matemàtica, Bellaterra, Spain; [3]MTA TTK NAP B Research Group of Sleep Oscillations, Budapest, Hungary; [4]Canadian Center for Behavioural Neuroscience, University of Lethbridge, Lethbridge, Canada

*For correspondence:
djercog@clinic.cat (DJ);
jrochav@clinic.cat (JR)

[†]These authors contributed equally to this work

Present address: [‡]Neurocentre Magendie, INSERM U1215, Bordeaux, France

Competing interests: The authors declare that no competing interests exist.

**Abstract** In the idling brain, neuronal circuits transition between periods of sustained firing (UP state) and quiescence (DOWN state), a pattern the mechanisms of which remain unclear. Here we analyzed spontaneous cortical population activity from anesthetized rats and found that UP and DOWN durations were highly variable and that population rates showed no significant decay during UP periods. We built a network rate model with excitatory (E) and inhibitory (I) populations exhibiting a novel bistable regime between a quiescent and an inhibition-stabilized state of arbitrarily low rate. Fluctuations triggered state transitions, while adaptation in E cells paradoxically caused a marginal decay of E-rate but a marked decay of I-rate in UP periods, a prediction that we validated experimentally. A spiking network implementation further predicted that DOWN-to-UP transitions must be caused by synchronous high-amplitude events. Our findings provide evidence of bistable cortical networks that exhibit non-rhythmic state transitions when the brain rests.
DOI: https://doi.org/10.7554/eLife.22425.001

## Introduction

A ubiquitous pattern of spontaneous cortical activity during synchronized brain states consists of the alternation between periods of tonic firing (UP states) and periods of quiescence (DOWN states) (*Luczak et al., 2007*; *Steriade et al., 1993a*; *Timofeev et al., 2001*). Cortical UP and DOWN dynamics take place during slow-wave-sleep (SWS) (*Steriade et al., 1993a*) and can also be induced by a number of anesthetics (*Steriade et al., 1993a*). More recently however, similar synchronous cortical dynamics have been observed not only in awake rodents during quiescence (*Luczak et al., 2007*; *Petersen et al., 2003*), but also in animals performing a perceptual task, both rodents (*Sachidhanandam et al., 2013*; *Vyazovskiy et al., 2011*) and monkeys (*Engel et al., 2016*).

Spontaneous activity resembling UP and DOWN states has been found in cortical slices in vitro (*Cossart et al., 2003*; *Fanselow and Connors, 2010*; *Mann et al., 2009*; *Sanchez-Vives and McCormick, 2000*), in slabs (*Timofeev et al., 2000*) and in vivo under extensive thalamic lesions (*Steriade et al., 1993b*). This suggests that the underlying mechanism has an intracortical origin. In such scenario, the standard hypothesis postulates that during UP periods a fatigue mechanism of cellular origin – e.g. spike frequency adaptation currents or synaptic short-term depression – decreases network excitability until the state of tonic firing can no longer be sustained and the cortical network switches into a DOWN state (*Contreras et al., 1996*; *Sanchez-Vives and McCormick, 2000*). During DOWN periods, in the absence of firing, the fatigue variables recover until the circuit becomes self-excitable and autonomously transitions into an UP state (*Cunningham et al., 2006*; *Le Bon-Jego and Yuste, 2007*; *Poskanzer and Yuste, 2011*; *Sanchez-Vives and McCormick, 2000*; *Timofeev et al., 2000*). This mechanism of activity dependent negative feedback causing oscillatory UP-DOWN dynamics has been implemented by several computational models (*Bazhenov et al.,*

*2002*; *Benita et al., 2012*; *Chen et al., 2012*; *Compte et al., 2003b*; *Ghorbani et al., 2012*; *Hill and Tononi, 2005*; *Parga and Abbott, 2007*). However, although commonly described as a slow oscillation, the rhythmicity of UP-DOWN dynamics has not been systematically quantified and seems to depend on the details of the preparation (*Chauvette et al., 2011*; *Erchova et al., 2002*; *Lampl et al., 1999*; *Ruiz-Mejias et al., 2011*).

Alternatively, there is strong evidence suggesting that UP-DOWN transitions in neocortical circuits are coupled with activity in subcortical and limbic areas. Thalamocortical neurons for instance can endogenously oscillate at low frequencies (*Hughes et al., 2002*; *McCormick and Pape, 1990*), cause cortical UP states when stimulated (*Rigas and Castro-Alamancos, 2007*) or modulate the UP-DOWN dynamics when suppressed (*David et al., 2013*; *Lemieux et al., 2014*) and their spontaneous activity correlates with UP state onset (*Contreras and Steriade, 1995*; *Slézia et al., 2011*; *Ushimaru et al., 2012*). Moreover, the timing of hippocampal sharp-wave ripples (*Battaglia et al., 2004*), or basal ganglia activity (*Ushimaru et al., 2012*) also tends to precede DOWN to UP transitions. Finally, intracortical stimulation can effectively cause UP-DOWN transitions (*Beltramo et al., 2013*; *Shu et al., 2003*) even when only a few dozen neurons are stimulated (*Stroh et al., 2013*). In total, these findings describe a system whose macroscopic UP-DOWN dynamics are sensitive to temporal fluctuations of both external inputs and local circuit activity. Such a network would in principle generate unpredictable and therefore irregular UP-DOWN dynamics, since transitions are no longer dependent exclusively on local cortical internal dynamics.

The interplay of fatigue mechanisms and fluctuations causing transitions between two states has been theoretically studied in the developing spinal cord (*Tabak et al., 2011*; *2000*), and in the context of UP-DOWN dynamics mostly in networks composed of excitatory units (*Holcman and Tsodyks, 2006*; *Lim and Rinzel, 2010*; *Mattia and Sanchez-Vives, 2012*; *Mejias et al., 2010*). Most models of spontaneous activity are however theoretically founded on the balance between excitatory (E) and inhibitory (I) populations (*Amit and Brunel, 1997*; *van Vreeswijk and Sompolinsky, 1998*), a dynamic state that can quantitatively mimic population spiking activity during desynchronized states (*Renart et al., 2010*). Analysis of cortical responses in the visual cortex suggest that cortical networks operate in the inhibition-stabilized regime, in which recurrent excitatory feedback alone is strong enough to destabilize the network activity but feedback inhibition maintains stability (*Ozeki et al., 2009*). In spite of growing evidence showing that the interaction between E and I populations is critical in generating spontaneous activity, the conditions under which an EI network model can exhibit a robust bistability between a low-rate inhibition-stabilized state and a quiescent state are still not well understood (*Latham et al., 2000*). To develop such a model, we first performed population recordings of ongoing cortical activity during synchronized brain state epochs in rats under urethane anesthesia (*Détári and Vanderwolf, 1987*; *Luczak et al., 2007*; *Murakami et al., 2005*; *Whitten et al., 2009*). Analysis of population single-unit spiking dynamics, showed irregular UP and DOWN periods and no decay of the average rate during UPs. Given these constraints, we built an EI network model that, capitalizing on the firing threshold non-linearity and the asymmetry of the E and I transfer functions, exhibited a novel type of bistability with a quiescent (DOWN) and a low-rate state (UP). External input fluctuations into the network caused the irregular UP-DOWN transitions. Adaptation in E cells in contrast, did not cause transitions and had a different effect on the E rate in each of the two states: while it exhibited recovery during DOWN periods, it showed almost no decay during UP periods due to the balanced nature of the UP dynamics. In addition, a spiking network implementation of the model revealed that external input fluctuations to neurons in the network cannot respond to simple independent Gaussian statistics but must include stochastic, synchronous high-amplitude events that can trigger DOWN-to-UP transitions. Our model provides the first EI network that exhibits stochastic transitions between a silent and a low rate inhibition-stabilized attractor matching the statistics of UP and DOWN periods and population rate time-courses observed in the cortex.

## Results

To investigate the mechanisms underlying the generation of spontaneous cortical activity, we recorded the spiking activity from large populations of neurons (mean ±SD = 64±23 cells) in deep layers of somatosensory cortex of urethane-anesthetized rats (n = 7 animals) (*Barthó et al., 2004*; *Luczak et al., 2009*). Because brain state under urethane can vary spontaneously (*Détári and*

*Vanderwolf, 1987*; *Luczak et al., 2007*; *Murakami et al., 2005*; *Whitten et al., 2009*), we selected the most clearly synchronized epochs characterized by the stable presence of high-amplitude, slow fluctuations in cortical local field potential (LFP) signals (*Figure 1A*; see Materials and methods) (*Harris and Thiele, 2011*; *Steriade et al., 2001*). During these epochs, the instantaneous population rate $R(t)$, calculated by merging all the recorded individual spike trains, displayed alternations between periods of tonic firing and periods of silence (*Luczak et al., 2007*), a signature of UP and DOWN states from an extracellular standpoint (*Figure 1B–C*) (*Cowan and Wilson, 1994*; *Sanchez-Vives and McCormick, 2000*; *Steriade et al., 1993a*). Despite the clear presence of UP and DOWN states, the population activity displayed no clear traces of rhythmicity as revealed by strongly damped oscillatory structure in both autocorrelograms of LFP and $R(t)$ (*Figure 1D and E*, respectively). Motivated by this, we hypothesized that the cortical circuit might transition between two network states in a random manner (*Deco et al., 2009*; *Mejias et al., 2010*; *Mochol et al., 2015*). Using a probabilistic hidden semi-Markov model (*Chen et al., 2009*), we inferred the instantaneous

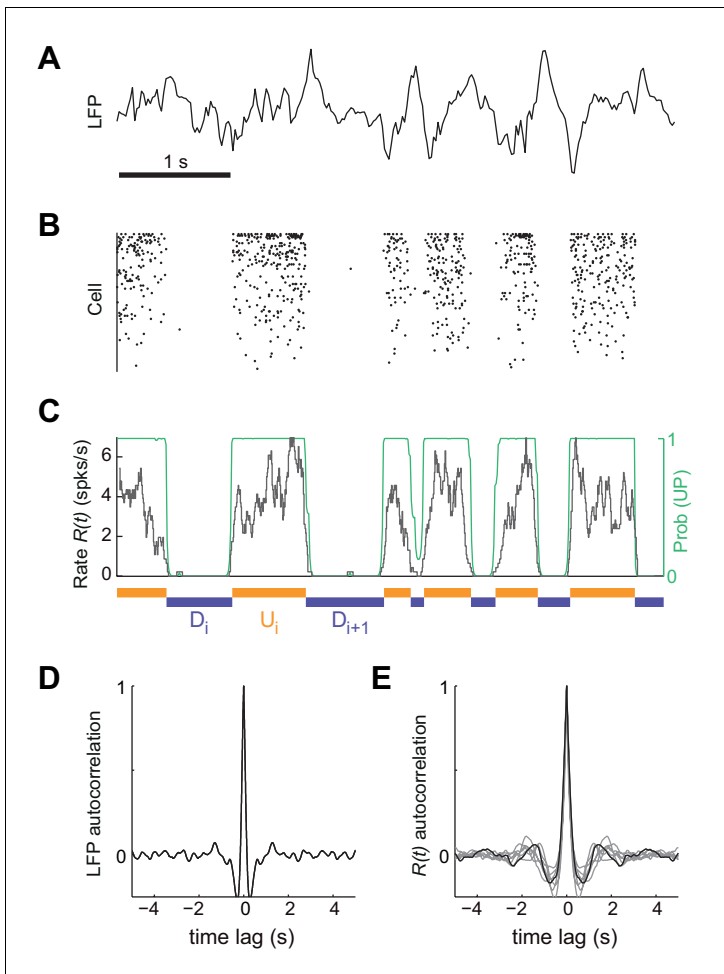

**Figure 1.** Synchronized brain activity under urethane anesthesia in the rat somatosensory cortex and the detection of putative UP and DOWN periods. (**A**) Local field potential during 5 s of synchronized state displaying high-amplitude slow fluctuations. (**B**) Population raster of 92 simultaneously recorded single units exhibiting the alternation between periods of tonic spiking activity and periods of neural quiescence (cells sorted based on mean firing rate). (**C**) Instantaneous population rate $R(t)$ (grey) is used to identify putative U (orange) and D (purple) periods. The detection algorithm is based on fitting a Hidden Markov Model (HMM) and computing the posterior probability of the hidden state being in an UP state (green) (see Materials and methods). (**D**) Average autocorrelogram of LFP (20 s windows) for one example experiment. (**E**) Average autocorrelogram of $R(t)$ for different (n = 7) experiments (example experiment in black).

DOI: https://doi.org/10.7554/eLife.22425.002

state of the circuit from the population rate $R(t)$ by extracting the sequence of putative UP (U) and DOWN (D) periods (*Figure 1C*, Materials and methods).

## UP and DOWN duration statistics during synchronized states

The statistics of U and D period durations showed skewed gamma-like distributions (*Figure 2A and B* right; *Figure 2—figure supplement 1*). The mean duration across different experiments displayed a wide range of values (*Figure 2B* left; mean ±SD:<U>= 0.43 ± 0.19 s, <D> = 0.46 ± 0.1 s, n = 7), whereas the coefficients of variation CV(U) and CV(D) of U and D periods, defined as the standard deviation divided by the mean of the period durations within experiments, were systematically high (*Figure 2B* middle, mean ±SD: CV(U) = 0.68 ± 0.09, CV(D) = 0.69 ± 0.1; median CV(U) = 0.64, CV(D) = 0.71). The irregularity in the U and D periods did not result from slow drifts in the mean U or D durations caused by variations of brain state as confirmed by computing the $CV_2$ (*Holt et al., 1996*), a local measure of irregularity that is less affected by slow variations in the statistics (mean ±SD: $CV_2(U)$ = 0.86 ± 0.13, $CV_2(D)$ = 0.75 ± 0.17; see Materials and methods). The high variability of U and D periods is consistent with the non-periodicity of the dynamics revealed in the autocorrelation function (*Figure 1D–E*).

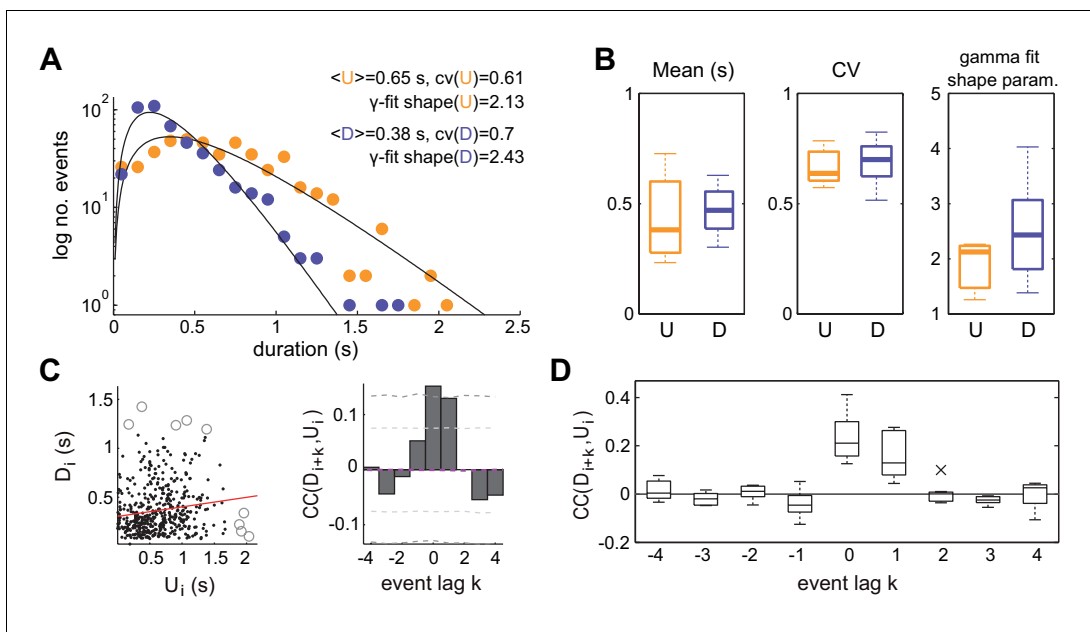

**Figure 2.** Statistics of U and D periods during synchronized brain activity. (**A**) Distribution of U and D durations for one example experiment (same as *Figure 1*). Inset shows the mean and coefficient of variation (CV) of U and D durations. (**B**) Summary of period duration mean (left), CV (middle), and gamma-fit shape parameter (right) across experiments (n = 7 rats). While average durations are quite heterogeneous across experiments, the period duration variability is consistently large. (**C**) Left: D duration ($D_i$) vs the consecutive U duration ($U_i$) exhibit weak but significant serial correlation. Values more than 3 standard deviations away from the mean (circles) were discarded for correlation analysis. Red line shows linear regression. Right: Cross-correlogram between the $D_i$ and $U_i$ sequences for different lags (**k**) in a single experiment. Magenta dashed line represent the mean cross-correlogram from a local shuffled (see Materials and methods). Light (dark) grey dashed line showing 95% C.I. point-wise (global) error bands. (**D**) Summary of cross-correlation analysis for the different experiments, displaying consistent positive correlations across experiments for lags k = 0 and k = 1.

DOI: https://doi.org/10.7554/eLife.22425.003

The following source data and figure supplement are available for figure 2:

**Source data 1.** U and D period durations and statistics for individual experiments.
DOI: https://doi.org/10.7554/eLife.22425.005

**Figure supplement 1.** Distributions of U and D period durations for individual experiments.
DOI: https://doi.org/10.7554/eLife.22425.004

We then asked whether the lengths of U and D periods were independent, as if the transitions between the two network states would reset the circuit's memory, or if in contrast they were correlated by a process impacting the variability of several consecutive periods. We computed the linear cross-correlation $Corr(U_i, D_{i+k})$ (*Figure 2C* left, for $k = 0$) between pairs of periods separated in the D-U sequence by a lag $k$ (*Figure 2C*, right). The cross-correlation $Corr(U_i, D_{i+k})$ displayed consistently non-zero values for k = 0 and k = 1 (mean ±SD: 0.21 ± 0.09, 0.17 ± 0.09, respectively; significant cross-correlation in 6/7 animals, p<0.05 permutation test), whereas it remained close to zero for the rest of lags, showing that period duration correlation is limited to adjacent periods (*Figure 2C–D*). The positive correlation between adjacent periods was not due to slow changes in their duration, as we corrected by the correlation obtained from surrogate D-U sequences obtained from shuffling the original sequence within 30 s windows (see Materials and methods). Positive correlations between consecutive periods of activity and silence can be generated when fluctuation driven transitions are combined with an adaptive process such as activity-dependent adaptation currents (*Lim and Rinzel, 2010*; *Tabak et al., 2000*): if a fluctuation terminates a U period prematurely without much build-up in adaptation, the consecutive D period also tends to be shorter as there is little adaptation to recover from. However, a major role of adaptation currents in dictating UP-DOWN dynamics (*Compte et al., 2003b*) seems at odds with the lack of rhythmicity and the highly variable U and D durations, indicative of a stochastic mechanism causing the transitions between network states.

## Spiking activity during UP and DOWN states

We next searched for more direct evidence of an adaptive process by examining the time course of the population firing rate $R(t)$ during U and D periods (see *Figure 1C*; see Materials and methods). The mean firing rate in U periods was low (mean ±SD: 3.72 ± 0.9 spikes/s, n = 7). Moreover, D periods displayed occasional spiking (mean ±SD rate 0.018 ± 0.007 spikes/s; see e.g. *Figure 3A–B* and *Figure 3—figure supplement 1*), in contrast with the idea that DOWN periods do not display spiking activity (*Chauvette et al., 2010*), but see (*Compte et al., 2003b*). Thus, our hypothesis was that adaptation currents, if present, would induce a decay in $R(t)$ during Us and an increase during Ds, and this impact on $R(t)$ dynamics should be more evident during longer periods due to a larger accumulation (during Us) or recovery (during Ds) of the adaptation. For each experiment, we aligned the rate $R(t)$ at the DOWN-to-UP (DU) and UP-to-DOWN (UD) transition times (*Figure 3A*). We then computed the average rates $R_{DU}(\tau)$ and $R_{UD}(\tau)$ across all DU and UD transitions, respectively, with $\tau = 0$ representing the transition time (*Figure 3B-C*; mean across experiments = 598 transitions; range 472–768). Because Us and Ds had different durations, we selected long periods (U, D > 0.5 s) and compared $R_{DU}(\tau)$ and $R_{UD}(\tau)$ at the beginning and end of each period (mean number of Us 181, range 61–307; Ds 202, range 55–331). To specifically assess a change in rate during the U period, we compared the average $R_{DU}(\tau)$ in the time window $\tau = (50, 200)$ ms (beginning of U) with the average $R_{UD}(\tau)$ in the window $\tau = (-200, -50)$ ms (end of U), which we referred to as U-onset and U-offset windows, respectively. The windows were chosen 50 ms away from $\tau = 0$ to avoid the transient change due to the state transitions (*Figure 3C–D*). We found no significant mean difference between population average rate at U-onset and U-offset windows across our experiments (mean ±SD onset minus offset population rate 0.04 ± 0.40 spikes/s, p=1, Wilcoxon signed rank, n = 7 animals). The equivalent analysis performed on D periods yielded a small but significant mean increase in the population rate between the D-onset and D-offset windows (mean ±SD −0.014 ± 0.013 spikes/s, p=0.047, Wilcoxon signed rank test). To examine in more detail the lack of population rate change during Us, we looked at the modulation of individual neuron rates normalized by the overall temporal average of each unit (*Figure 3E*). We found that the change between U-onset and U-offset averaged across all our neurons (n = 448 cells) was not significantly different from zero (*Figure 3E* right, mean ±SD of the onset vs offset difference of normalized rates 0.057 ± 1.163, p=0.12, Wilcoxon signed rank test) but that the recovery during D periods was significant (*Figure 3E* left; mean ±SD −0.015 ± 0.087, p=0.0002, Wilcoxon signed rank test). Some individual neurons however did show a significant modulation between U-onset and U-offset, but the decrease found in a fraction of the neurons was compensated with a comparable increase in another fraction of neurons (*Figure 3E* right). Thus, at the population level, spiking activity during U periods displayed a sustained time course with no significant traces of rate adaptation.

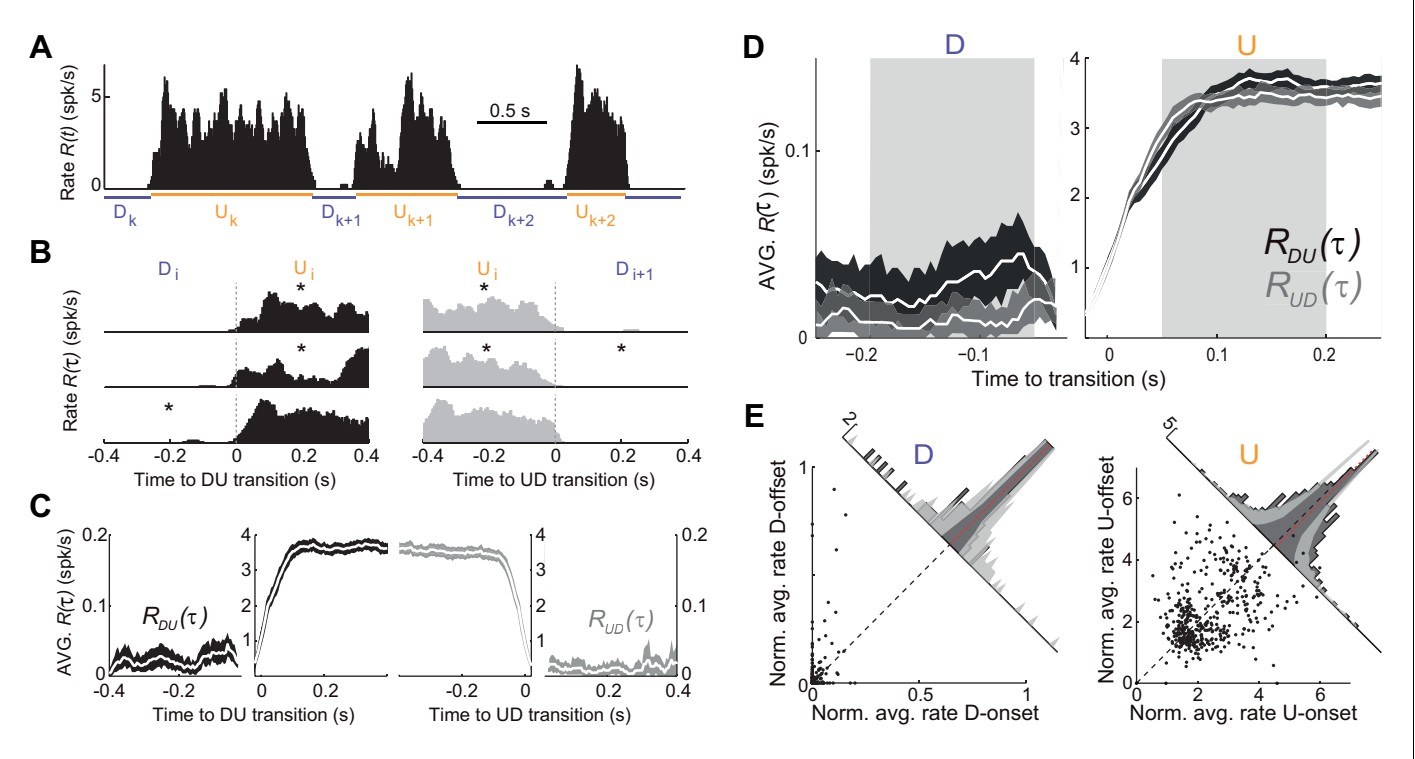

**Figure 3.** Population spiking statistics during U and D periods. (**A**) Example of instantaneous population rate $R(t)$ with U and D detected periods (as in *Figure 1*). (**B–C**) Each U period is aligned at the DU (B, left) and UD (B, right) transition times in order to compute the instantaneous population rate averaged across transitions $R_{DU}(\tau)$ (C, dark grey) and $R_{UD}(\tau)$ (C, light grey), respectively. Only periods longer than 0.5 s (asterisks in B) were included in the average. (**D**) Comparison of population rate at the onset and offset of Us and Ds done by overlaying $R_{DU}(\tau)$ and a time-reversed $R_{UD}(\tau)$. Onset and offset windows defined during D and U periods (shaded) were used to test significance of changes in the rate. (**E**) Normalized firing rates from all individual neurons (448 cells from n = 7 animals) during onset and offset windows. Left: D periods. Right: U periods. Average across cells is shown in red. Gray bands show 95% C.I. of the histograms obtained from onset-offset shuffled data (see Materials and methods).
DOI: https://doi.org/10.7554/eLife.22425.006

The following source data and figure supplement are available for figure 3:

**Source data 1.** Instantaneous population rate averaged across transitions $R_{DU}(\tau)$ and $R_{UD}(\tau)$ for individual experiments.
DOI: https://doi.org/10.7554/eLife.22425.008

**Figure supplement 1.** Firing rate statistics for single units during U and D periods.
DOI: https://doi.org/10.7554/eLife.22425.007

## Rate model for UP and DOWN dynamics

To understand the network and cellular mechanisms underlying the generation of stochastic U-D dynamics, showing U-D serial correlations and sustained rates during U periods, we analyzed a computational rate model composed of an excitatory (E) population recurrently coupled with an inhibitory (I) population (*Latham et al., 2000*; *Ozeki et al., 2009*; *Tsodyks et al., 1997*; *Wilson and Cowan, 1972*). The excitatory-inhibitory (EI) network model described the dynamics of the mean instantaneous rates $r_E$ and $r_I$ of each population in the presence of fluctuating external inputs. In addition, the E population included an adaptation mechanism, an additive hyperpolarizing current $a$ that grew linearly with the rate $r_E$ (*Figure 4A*; see Materials and methods). We did not consider adaptation in the inhibitory population for simplicity, and because inhibitory neurons show little or no spike-frequency adaptation when depolarized with injected current (*McCormick et al., 1985*). Our aim was to search for a regime in which, in the absence of adaptation and external input fluctuations, the network exhibited bistability between a quiescent (D) and a low-rate state (U) fixed point. Although bistability in low-dimensional EI networks has been described since the seminal work of *Wilson and Cowan (1972)*, previous models primarily sought to explain bistability between a low-rate and a high-rate state, and exploited the combination of expansive and contractive non-

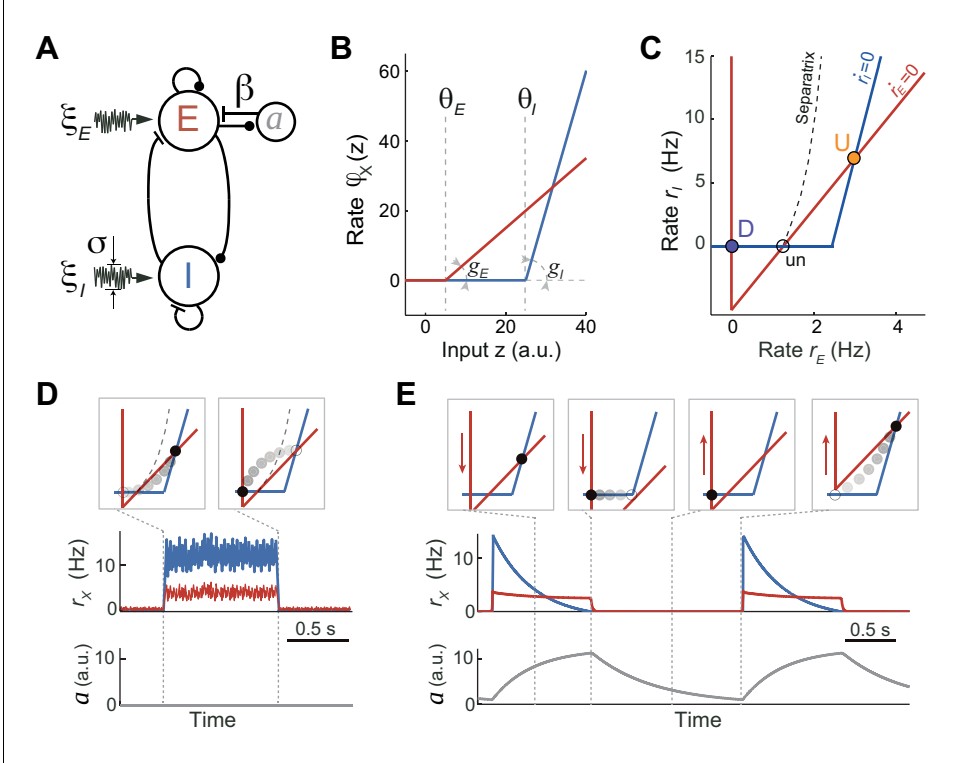

**Figure 4.** Rate model for fluctuations and adaptation induced UP and DOWN dynamics. (**A**) Network composed of recurrently connected inhibitory (I, blue) and excitatory (E, red) populations, with E exhibiting rate adaptation $a(t)$ and both populations receiving independent fluctuating external inputs. (**B**) Transfer functions for the E and I populations are threshold-linear with unequal thresholds $\theta_E < \theta_I$ and unequal gains $g_E < g_I$. This marked asymmetry is at the origin of the bistability obtained in the network. (**C**) In the absence of adaptation, the phase plane of rates $r_E$ vs. $r_I$ shows the E and I nullclines (red and blue, respectively) whose intersections determine two stable (**U** and **D**) and one unstable (**un**) fixed points. The separatrix (dashed line) divides the phase plane into the basins of attraction of the D and U stable points. (**D, E**) Schematics of fluctuations-induced DU and UD transitions in the absence of adaptation ($\beta = 0$) and adaptation-induced transitions in the absence of fluctuations ($\sigma = 0$), respectively. Traces of $r_E(t)$, $r_I(t)$ and adaptation $a(t)$ illustrate steady fluctuating rates during U periods when there is no adaptation (**D**), and a periodic alternation between U and D characterized by a strongly decaying I rate during Us when there is no fluctuations (**E**). Top insets show the network trajectories in the phase-plane taken at different time points (vertical dotted lines). Notice the downward (upward) displacement of the E-nullcline during U (D) periods (red arrows in E). Connectivity parameters: $J_{EE} = 5$, $J_{EI} = 1$, $J_{IE} = 10$, $J_{II} = 0.5$ s; Transfer function parameters: $g_E = 1$, $g_I = 4$ Hz, $\theta_E = 0$, $\theta_I = 25$ a.u.

DOI: https://doi.org/10.7554/eLife.22425.009

linearities produced by the transfer function (**Amit and Brunel, 1997**; **Renart et al., 2007**; **Wilson and Cowan, 1972**), short-term synaptic plasticity (**Hansel and Mato, 2013**; **Mongillo et al., 2008**) or the divisive effect of inhibitory conductances (**Latham et al., 2000**) (see Discussion). We found that the expansive nonlinearity of the transfer function alone was sufficient to obtain bistability between D and U states. Given this, we chose the simplest possible transfer function with a threshold: a threshold-linear function (**Figure 4B**, see Materials and methods). Our choice to only use an expansive threshold non-linearity constrained strongly the way in which the network could exhibit bistability as can be deduced by plotting the nullclines of the rates $r_E$ and $r_I$ (**Figure 4C**): only when the I nullcline was shifted to the right and had a larger slope than the E nullcline, the system exhibited two stable attractors (**Equation 20** in Materials and methods). This configuration of the nullclines was readily obtained by setting the threshold and the gain of the I transfer function larger than those of the E transfer function (**Figure 4B**), a distinctive feature previously reported when intracellularly characterizing the $f$ - $I$ curve of pyramidal and fast spiking interneurons in the absence of background synaptic activity (**Cruikshank et al., 2007**; **Schiff and Reyes, 2012**). This difference in

gains and thresholds in the E and I populations was not a necessary condition to obtain the bistability: alternatively, a proper selection of connectivity parameters with identical E and I transfer functions could satisfy the conditions to obtain similar bistable function (see Materials and methods, *Equations [20-22]*). This novel bistable regime yielded a quiescent D state, and arbitrarily low firing rates for both E and I populations during U states, depending on the values of the thresholds and the synaptic weights (*Figure 4C*). This is remarkable as in most bistable network models the rate of the sustained activity state is constrained to be above certain lower bound (see Discussion). Moreover, in this bistable regime, the U state is an inhibition-stabilized state, a network dynamical condition in which the excitatory feedback is so strong that would alone be unstable, but is balanced with fast and strong inhibitory feedback to maintain the rates stable (*Ozeki et al., 2009*; *Tsodyks et al., 1997*) (see Materials and methods).

There are two ways in which transitions between U and D states can occur. On the one hand, transitions could be driven by external input fluctuations, which were modeled as a stochastic process with zero mean and short time constant (*Figure 4D*). This fluctuating input reflected either afferents coming from other brain areas whose neuronal activity was stochastic and uncorrelated with the cortical circuit internal dynamics or the stochasticity of the spiking happening during U periods which was not captured by the dynamics of the rates (*Holcman and Tsodyks, 2006*; *Lim and Rinzel, 2010*). On the other hand, in the absence of fluctuations, state transitions could also occur solely driven by adaptation currents (*Figure 4E*). Because the adaptation time constant was much longer than the time constants of the E and I rates, the dynamics of the rates $r_E(t)$ and $r_I(t)$ relaxing rapidly to their steady-state can be decoupled from the slow changes in $a(t)$ (*Latham et al., 2000*; *Rinzel and Lee, 1987*). The network dynamics can be described in the phase plane ($r_E(t)$, $r_I(t)$) with variations in $a(t)$ causing a displacement of the E-nullcline. In particular, during U periods the build-up in adaptation produced a downward displacement of the E-nullcline (*Figure 4E*). If adaptation strength β was sufficiently large the displacement increased until the U state was no longer a fixed point and the network transitioned to the only stable fixed point D. Recovery of adaptation during D periods shifted the E-nullcline upwards until the D state disappeared and there was a transition to the U state (*Figure 4E*). In this limit cycle regime the network exhibited an oscillatory behavior with a frequency close to the inverse of the adaptation recovery time constant. When the two types of transitions are combined, two types of stability in U and D states can be distinguished: (1) metastable, referred to a state that was stable to the dynamics of both the rates and the adaptation but could transition away due to input fluctuations; (2) quasi-stable, referred to a state that was stable for the fast rate dynamics but unstable for the slow adaptation dynamics, plus it was also susceptible to fluctuation-driven transitions.

## UP and DOWN state statistics in the model

To quantify the relative impact of activity fluctuations and adaptation in causing U-D transitions in the data, we compared the dynamics of the model for different adaptation strengths β and different values of the E-cell effective threshold $\theta_E$ (defined as the difference between the activation threshold and the mean external current). The ($\theta_E$,β) plane was divided into four regions with UD alternations, corresponding to the four combinations of metastability and quasi-stability (*Figure 5A*). Since only metastable states tend to give exponentially distributed durations with CV ~1, the large variability found in both U and D durations (*Figure 2B*) constrained the model to the subregion where both states were metastable and UD and DU transitions were driven by fluctuations (red area in *Figure 5A*). The existence of serial correlations between consecutive U and D in the data (*Figure 2C–D*) discarded an adaptation-free regime (β = 0), in which transitions were solely driven by fluctuations and the duration of each period was independent of previous durations (*Figure 5B* right). Thus, we explored a regime with β >0 but still in the region where both states were metastable (*Figure 5B*, green square) and the input fluctuations produced alternation dynamics (*Figure 5C* top) with broad U and D duration distributions and relatively high CVs (*Figure 5D* top). The magnitude of the fluctuations was adjusted to obtain frequent transitions in this region and serial correlations quantitatively comparable with the data (*Figure 5—figure supplement 1*). Moreover, the rates showed an autocorrelation function qualitatively similar to the data, with negative side-lobes but no clear traces of rhythmicity (*Figure 5E*). Adaptation introduced correlations across consecutive periods (*Figure 5D* bottom) because at the transition times the system kept a memory of the previous period in the adaptation value $a(t)$. For adaptation to introduce substantial correlations, $a(t)$ had to

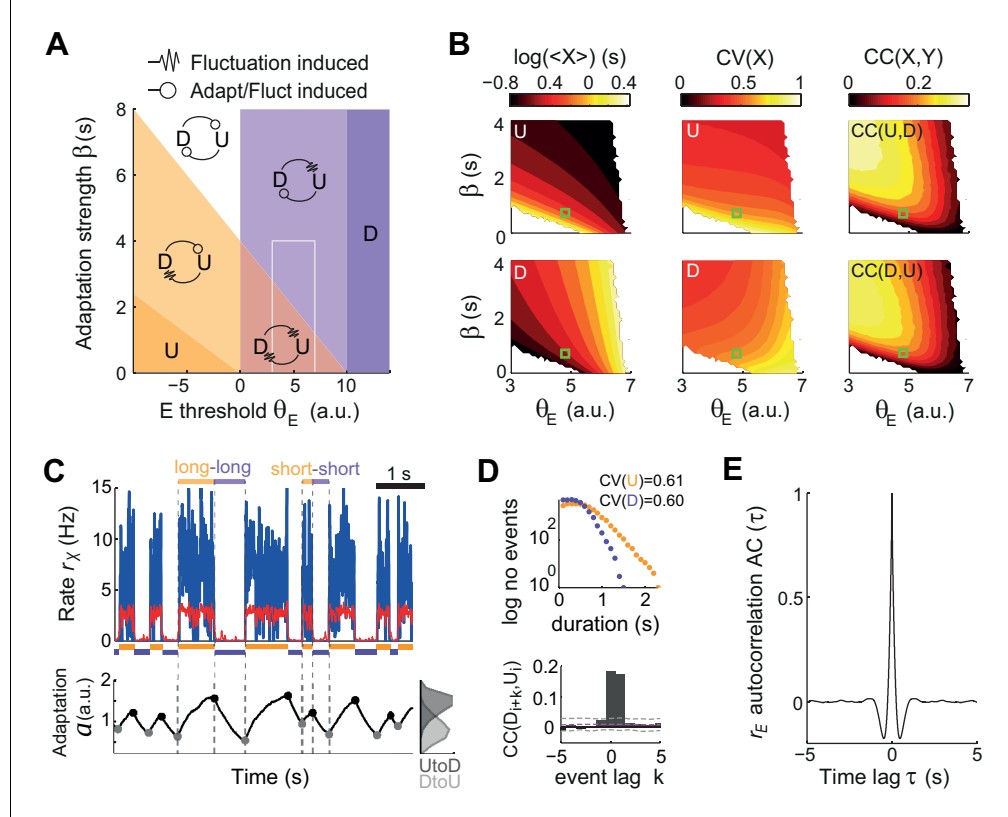

**Figure 5.** Fluctuations and weak adaptation are required in the model to explain the U-D statistics of the data. (**A**) Different dynamical regimes of the model as a function of the adaptation strength $\beta$ and the effective threshold $\theta_E$. Each U and D state is either meta-stable or quasi-stable depending on whether the transitions to the opposite state can be caused by fluctuations or adaptation + fluctuations, respectively (see arrow code in top inset). There are five region types: regions with a single stable state and no transitions (dark purple and dark orange), a region with both U and D meta-stable (light red), one with both U and D quasi-stable (white) and mixed regions with a meta-stable and a quasi-stable state (light orange and light purple). (**B**) Statistics of U (top) and D (bottom) periods obtained from numerical simulations: mean durations (left), duration CV (center) and of cross-correlation CC of consecutive periods (right) as a function of $\beta$ and $\theta_E$. The region analyzed is marked in A (gray rectangle). Fluctuations were $\sigma$ = 3.5. White areas indicate very low transition rate. (**C–E**) Model example quantitatively reproducing some U-D statistics of the data. The $\beta$ and $\theta_E$ used are marked in B (green square; $\theta_E$ = 4.8 a.u., $\beta$ = 0.7 Hz$^{-1}$). Example traces of $r_E(t)$, $r_I(t)$, and $a(t)$ show U-D transitions with irregular durations (C). Black and gray filled dots indicate the adaptation values at the UD and DU transition times, respectively. The corresponding histograms illustrate the variability of these values (C bottom right). (**D**) Top: Distributions of U and D period durations. Bottom: Cross-correlograms of D and U periods for different lag values (compare with **Figure 2C**). Grey dashed lines show global error bands and magenta dashed line shows mean CC of shuffles. (**E**) Autocorrelogram of $r_E(t)$ shows no traces of rhythmicity.

DOI: https://doi.org/10.7554/eLife.22425.010

The following figure supplement is available for figure 5:

**Figure supplement 1.** Statistics of U and D durations as a function of the adaptation strength $\beta$ and the effective threshold $\theta_E$ for different amplitude of fluctuating external inputs.

DOI: https://doi.org/10.7554/eLife.22425.011

be variable at the transition times (**Lim and Rinzel, 2010**), a condition that required adaptation to be fast, to vary within one period, but not too fast to prevent reaching the equilibrium (**Figure 5C** bottom trace). Thus, when a strong fluctuation caused a premature UD transition, i.e. a short $U_k$, adaptation had no time to build up and tended to be small, increasing the probability of a premature DU transition in the following D period, i.e. a short $D_{k+1}$. Conversely, a long $U_k$ recruited strong adaptation that required a long $D_{k+1}$ to recover (see highlighted examples in **Figure 5C**). In this

regime, the dynamics of adaptation *a(t)* alone did not cause transitions but did strongly modulate the probability that an external fluctuation would cause a transition (*Moreno-Bote et al., 2007*). Altogether, this analysis suggests that the observed U-D dynamics occurred in a regime with strong random fluctuations, that these fluctuations were necessary to cause the transitions, and that adaptation modulated the timing of the transitions and consequently introduced correlations between the duration of consecutive periods.

## Dynamics of E and I populations during UP and DOWN states: model and data

According to the model, adaptation currents in the E population can parsimoniously account for the U-D serial correlations but this is in apparent contradiction with the fact that the population rate $R(t)$ in the data did not decrease significantly during U periods (*Figure 3C–E*). To reconcile these two seemingly contradictory observations we used the model with the parameters that matched the data's U and D statistics (*Figure 5C–E*) to characterize the time course of the rates $r_E(t)$ and $r_I(t)$ averaged across DU and UD transitions. Interestingly, the average $r_E(t)$ at the beginning and at the end of U periods did not show much difference whereas the average $r_I(t)$ showed a larger decrease over the U period (*Figure 6A*). Thus, although only the E and not the I population included intrinsic adaptation mechanisms, it was $r_I(t)$ the one that exhibited the most pronounced decay during U periods. This was a direct consequence of the specific conditions that gave rise to bistability in our model: the difference in thresholds, that is, $\theta_I > \theta_E$, and the fact that the I-nullcline has a higher slope than the E-nullcline (*Equation 21* in Materials and methods). These features imposed that as adaptation built up during U periods, the downward displacement of the E-nullcline caused a greater decrease in $r_I(t)$ than in $r_E(t)$ (compare 'decay' colored bands in *Figure 6B*). With this arrangement the drop in $r_E(t)$ could be made arbitrarily small by increasing the slope of the I-nullcline (*Figure 6B*). Note that

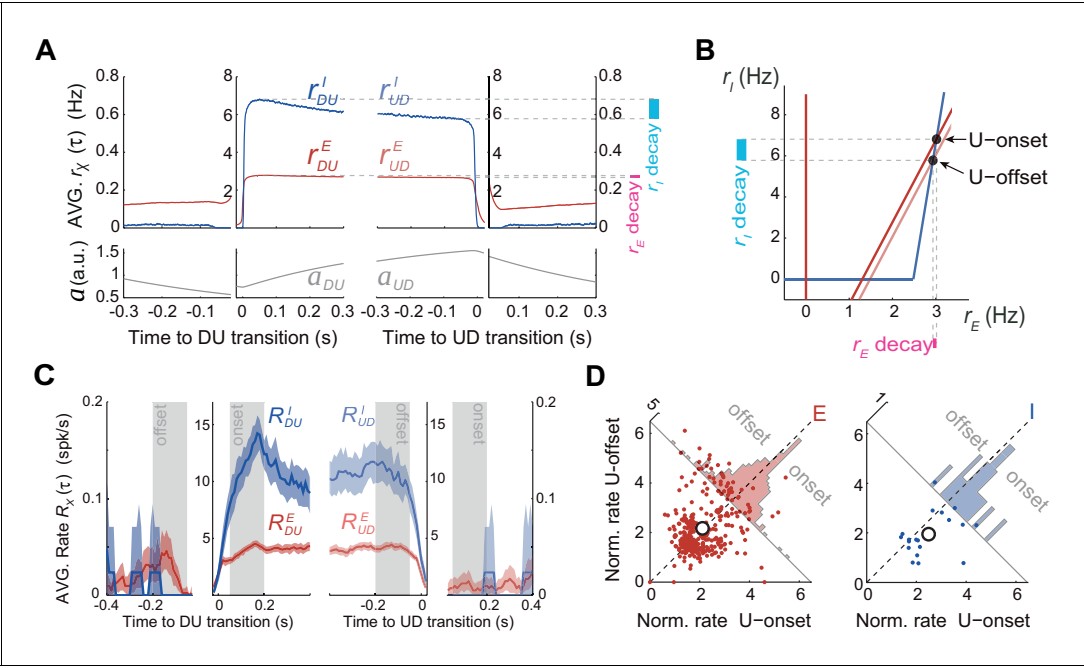

**Figure 6.** Excitatory and inhibitory populations during UP and DOWN alternation dynamics. (A) Model average population rates $r_E$ and $r_I$ and adaptation $a$ as a function of time, aligned at DU and UD transitions (same simulation parameters as in *Figure 5C*). (B) Model predicts a pronounced decay for $r_I$ (cyan bar) with minimal decay of $r_E$ (pink bar) throughout UP periods, despite adaptation is exclusively included in E cells (*Figure 4A*). (C) Example experiment averaged putative excitatory and inhibitory population rates ($R^E(\tau)$ and $R^I(\tau)$, respectively) aligned at DU and UD transitions. (D) Normalized firing rates from individual neurons (see Materials and methods) pooled from different experiments (n = 5; 330 putative E cells and 21 putative I cells active during U) comparing the activity from putative E and I cells during U onset and offset periods (gray shaded areas from panel C), reveals a significant decrease of I cells during U periods.

DOI: https://doi.org/10.7554/eLife.22425.012

this feature of the model is not dependent on its specific regime of operation, as it would similarly apply in an adaptation-driven regime (*Figure 4E*). During D periods the average $r_E(t)$ did show a substantial increase due to the recovery of adaptation, whereas the $r_I(t)$ did not. This was because in the D state, the quiescent network behaved as isolated neurons reflecting the dynamics of intrinsic adaptation which was only present on E cells. In sum, if the majority of the neurons that we recorded experimentally were excitatory, the model could explain why adaptation currents did not cause a significant decrease in the average rate during U periods (*Figure 3C–D*). The model in addition predicts that the rate of inhibitory neurons should exhibit a noticeable decrease during U periods.

Motivated by this prediction, we investigated the dynamics of the rates of excitatory and inhibitory neurons during U and D periods in the experimental data. Based on spike waveforms, isolated units from n = 5 experiments were classified into putative interneurons (I) and putative excitatory neurons (E), following previously described procedures (*Barthó et al., 2004*). The average rate for E and I populations ($R_E(t)$ and $R_I(t)$, respectively) displayed similar profiles across UD alternations, although higher values were observed for I cells during Us (see example experiment in *Figure 6C*). To assess the modulation of the rates during U periods, we looked at the normalized individual rates of all the E and I neurons (n = 330 and 21, respectively). As predicted by the model (*Figure 6A-B*), I cells displayed a significant rate decay during U periods that was not observed in E cells (*Figure 6D*; mixed-effects ANOVA with factors neuron type (E/I), onset/offset and neuron identity and experiment as random factors: interaction neuron type x onset/offset $F(1,349)=6.3$, p=0.013). During D periods, E cells also showed a significant increase in rate (Wilcoxon signed rank test p=0.0092), just like that observed in the whole cell population, whereas no rate change was found in I cells (not shown). Although these changes observed during D periods were also predicted by the model, properly testing the significance of this interaction would require a larger data set with more I cells. The validation of the prediction on the counter-intuitive emergent dynamics of E and I rates during U periods strongly suggests that the mechanism dissected by the model underlies the putative bistability observed in cortical circuit dynamics.

## Dynamics of state transitions in a spiking EI network

To assess whether the mechanism for state transitions proposed by the rate model could generate UP-DOWN dynamics in a more biophysically realistic circuit we built a network composed of $N_E$ = 4000 excitatory and $N_I$ = 1000 inhibitory leaky integrate-and-fire spiking units (*Ricciardi, 1977*) (all-to-all connectivity). We used current-based synapses (*Brunel and Sergi, 1998*) and introduced a spike-based after-hyperpolarization (AHP) current in E cells (*Wang, 1998*; *La Camera et al., 2004*). The EI asymmetry in spike threshold and $f$ - $I$ gain described in the rate model was implemented and, using standard mean-field methods (*Amit and Tsodyks, 1991*), we revealed the same network bistability described above (compare *Figures 7A* and *4C*): a saddle node bifurcation gave rise to a quiescent branch (DOWN) co-existing with a low-rate branch (UP; *Figure 7B*). Numerical simulations showed that while the network was in the UP state the AHP current increased moving the system along the upper branch towards the saddle-node and just causing a small decrease in $r_E$. However, because we chose a small AHP amplitude so that the network operated in the bistable regime (see fixed points in *Figure 7B*), the adaptation buildup alone did not trigger an UP to DOWN transition. It was the current fluctuations produced by the irregular activity during UP periods that triggered UP to DOWN transitions. However, once the network was in the DOWN state the external independent Gaussian inputs only caused subthreshold membrane fluctuations in E cells that sat far away from the spiking threshold (voltage std. dev. 2.5 mV with distance from resting voltage to threshold of 12.4 mV). In these conditions, there was no spiking activity during DOWN periods and the network could not transition to the UP state (not shown). To make these subthreshold fluctuations effective in driving transitions, we first depolarized neurons so that their resting potential during the D state was closer to threshold and the recovery from adaptation alone was almost sufficient to cause the transitions (*Figure 7—figure supplement 2A–E*). This *ad hoc* depolarization was sufficient to generate UP-DOWN alternations but prevented the membrane potential from showing bi-modality (*Figure 7—figure supplement 2F*), the intracellular signature of UP and DOWN states. Moreover, the alternations had a very small serial correlation between consecutive D and U periods, Corr(D,U), (*Figure 7—figure supplement 2I*) because adaptation at the time of the DOWN-to-UP transition was narrowly distributed, did not retain information about the length of the DOWN period, and could thus not constrain the duration of the following U period (*Lim and Rinzel, 2010*).

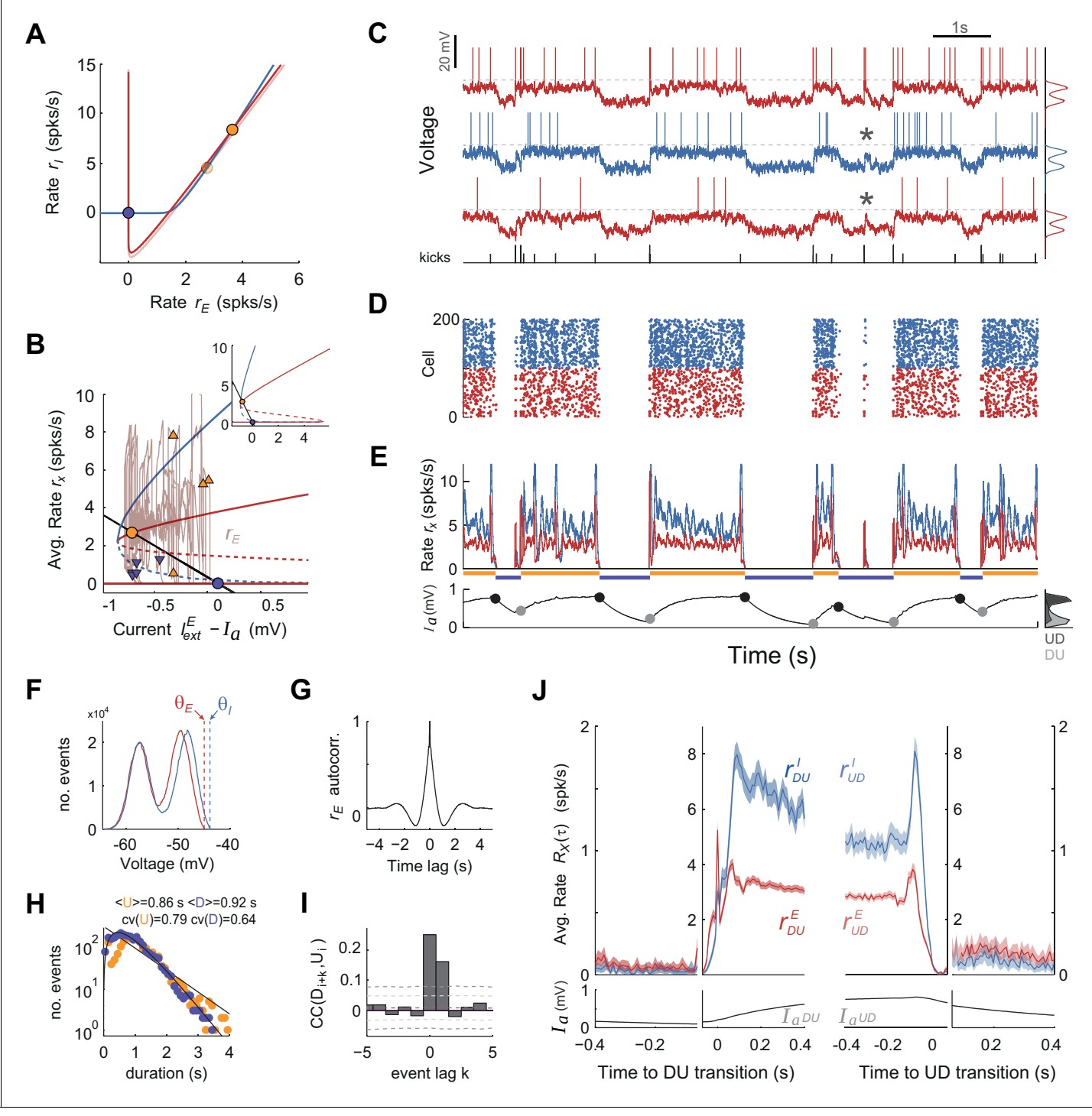

**Figure 7.** UP and DOWN dynamics in an EI network model of spiking neurons. (A) Phase plane of population averaged rates $r_E$ vs. $r_I$ showing the E and I nullclines (red and blue, respectively) obtained using mean-field expressions for the rates. The E nullcline is drawn twice assuming an AHP current $I_a$ fixed at the mean value observed at the UP onset (dark red) and offset (light red). Filled circles display UP (orange) and DOWN (purple) stable fixed points for each case. (B) Bifurcation diagram showing the stable (solid) and unstable (dashed) fixed points of the rates $r_E$ (red) and $r_I$ (blue) as a function of the difference between the external current $I^E_{ext}$ and $I_a$. The straight line shows the dependence of the AHP current at equilibrium with the rate $r_E$. Solid dots show fixed stable points of the system. Superimposed $(r_E, I_a)$ trace shows a 10 s example of UP to DOWN transitions obtained from simulations (also shown in C-E). Arrowheads mark the point in this trace where DU (orange) and UD (purple) transitions were detected. Note the considerable variance along the $I_a$ axes for the two sets of arrowheads, required to produce serial correlations. Inset shows a zoom-out of this plot where the two bifurcation points (saddle-nodes) are visible. (C–E) Network activity snapshot (duration 10 s) showing the membrane voltage of three

*Figure 7 continued on next page*

*Figure 7 continued*

example I (blue) and E (red) neurons (only the neuron in the top receives kicks), train of external kicks (tick size represents kick amplitude) (C), spike rastergram of 100 E and 100 I cells (D), population averaged rates $r_E(t)$ vs. $r_I(t)$ (E top) and population averaged AHP current $I_a(t)$ (E bottom). Orange and purple horizontal lines indicate U and D intervals detected automatically as with the experimental data (compare with *Figure 1*). (F) Membrane voltage distributions for E and I cells (top and middle neurons shown in C). Vertical dashed lines display their spiking thresholds $\theta_E$ and $\theta_I$. (G) Autocorrelogram of the population averaged rate $r_E(t)$. (H) Distribution of U and D durations (dots) and gamma fits (lines). Legend shows the mean and CV of U and D durations (order parameter of the fits were $\gamma_U$ = 1.1;0.7 and $\gamma_D$ = 2.3;0.4 - shape;scale parameters). (I) Cross-correlogram of D and U periods for different lag values (compare with *Figure 2C*). Light (dark) grey dashed lines show 95% C.I. point-wise (global) error bands. (J) Population averaged rates $r_E$ and $r_I$ (top) and AHP current $I_a$ (bottom) as a function of time, aligned at DU and UD transitions (as in *Figure 6A,C*).

DOI: https://doi.org/10.7554/eLife.22425.013

The following figure supplements are available for figure 7:

**Figure supplement 1.** Kicks impinging on a minority of neurons induce coherent network transitions, with effectiveness depending on AHP current values.

DOI: https://doi.org/10.7554/eLife.22425.014

**Figure supplement 2.** UP and DOWN dynamics in the EI spiking network model caused by independent Gaussian noise.

DOI: https://doi.org/10.7554/eLife.22425.015

We thus reasoned that the most parsimonious way to cause a DOWN to UP transition without disrupting the bi-modality of the membrane potential was to maintain resting neurons hyperpolarized and to introduce stochastic brief external excitatory synchronous inputs that caused large amplitude depolarizing voltage bumps in a targeted subpopulation of E and I neurons (*DeWeese and Zador, 2006*). The statistics of occurrence times of these bumps were Poisson (*Tan et al., 2013*) and their frequency (~2–3 Hz) and amplitude (~10–15 mV) were set so that (i) they could cause DOWN to UP transitions (*Figure 7C–E*) with non-rhythmic structure (*Figure 7G*) and yield U and D interval distributions similar to the data (*Figure 7H*) and (ii) the effectiveness in causing a transition was not all-or-none but depended on the population average AHP current amplitude $I_a$ (*Figure 7—figure supplement 1E*). This dependence occurs because the distance to the saddle point limiting the basin of attraction of the DOWN state decreases with $I_a$ (*Figure 7B*, red dashed line). This meant that some kicks during DOWN periods failed to cause a transition, giving rise to sporadic sparse firing during DOWN periods also seen as large bumps in the membrane potential of the targeted cells (see asterisks in *Figure 7C*; see Discussion). Because of this dependence of the transition probability on the recovery of the AHP current, the transition dynamics displayed serial correlations Corr(D, U) (*Figure 7I*), as observed in the data (*Figure 2C-D*). Once in the UP state, the external kicks caused an excess of excitatory and especially inhibitory activity that destabilized the UP state and generated a transition to the DOWN state. Because the effectiveness of the kicks causing these transitions also depended on the average AHP current (*Figure 7—figure supplement 1F*), there were significant serial correlations Corr(U,D) quantitatively comparable to the data (*Figure 7I*). As explained by the rate model, the averaged population rate $r_E(t)$ showed very weak decrease along the UP period whereas $r_I(t)$ decayed much more strongly, until the overshoot caused by kicks at the UP offset (*Figure 7J*). Neurons displayed a bi-modal distribution of the membrane potential (*Figure 7F*; *Figure 7—figure supplement 1E*) and during UP periods they fired low rate irregular spike trains (CV of the Inter-spike-interval was 0.74 for E cells and 0.94 for I cells). In sum, a spiking network model was able to reproduce the results described in the rate model given there exists a mechanism to generate stochastic, synchronous large-amplitude bumps.

## Discussion

Using cortical population recordings we have shown that UP and DOWN period durations are irregular and show positive serial correlation, but there is no significant decrease of population rate during UP periods. These findings seem inconsistent with one another, as some support, while other challenge the idea that UP-DOWN dynamics are caused by cell or synaptic adaptive mechanisms. Using a standard EI rate model network, we have proposed a novel bistable regime based only on the expansive threshold non-linearity of the transfer function and on a reported difference between E and I spiking thresholds. While fluctuations produce transitions between the quiescent state (D) and the inhibition-stabilized state of arbitrarily low rate (U), adaptation acting on the E population

facilitates the effect of fluctuations causing the transitions. Paradoxically, because of the difference in E and I thresholds, adaptation causes a marginal decay of E rates but a significant decay of I rates during UP periods. This counterintuitive prediction, specific to our model, was validated in the experimental data.

Adaptive processes constitute the mechanistic hallmark for the generation of cortical UP and DOWN dynamics (*Contreras et al., 1996*; *Sanchez-Vives and McCormick, 2000*; *Timofeev et al., 2000*). This principle has been used in several computational models, by implementing synaptic short-term depression (*Bazhenov et al., 2002*; *Benita et al., 2012*; *Ghorbani et al., 2012*; *Hill and Tononi, 2005*; *Holcman and Tsodyks, 2006*; *Mejias et al., 2010*), or activity-dependent adaptation currents (*Compte et al., 2003b*; *Destexhe, 2009*; *Latham et al., 2000*; *Mattia and Sanchez-Vives, 2012*). Consistent with an adaptive process generating the dynamics, UP and DOWN states observed in vitro display clear rhythmicity with Gaussian shaped UP and DOWN duration distributions (*Mattia and Sanchez-Vives, 2012*). An in vivo study using ketamine anesthesia in mice reported reduced UP and DOWN duration variability across multiple cortical areas with CVs around 0.2–0.4 (*Ruiz-Mejias et al., 2011*). Moreover, a comparison of the UP and DOWN dynamics in the cat observed under ketamine anesthesia and those found in slow wave sleep (SWS) showed that the alternations were more rhythmic under ketamine (*Chauvette et al., 2011*). In contrast, our data displayed large variability (mean CV(U)~CV(D) $\simeq$0.7) and skewed distributions of UP and DOWN period durations (*Figure 2B*), in agreement with previous studies using urethane anesthesia (*Dao Duc et al., 2015*; *Stern et al., 1997*). Although a direct comparison between the UP-DOWN dynamics under urethane anesthesia and during natural sleep has not been made, urethane seems to mimic sleep in several aspects. First, it induces spontaneous alternations between synchronized and desynchronized states (*Curto et al., 2009*; *Steriade et al., 1994*), resembling the alternations between SWS and REM sleep (*Clement et al., 2008*; *Whitten et al., 2009*). Second, the irregular UP-DOWN transitions observed under urethane anesthesia seem to resemble the variability observed in SWS (*Ji and Wilson, 2007*; *Johnson et al., 2010*). Preliminary analysis of rat and mouse prefrontal cortex during SWS with the same population-based U-D detection methods used here (Materials and methods) showed that U periods had comparable mean length but were more irregular (CV ~1) than under urethane anesthesia (*Figure 1B*) whereas D periods were shorter (mean ~150 ms) and slightly more regular (CV ~0.5) (unpublished observations). Such an asymmetry in the duration and irregularity of U-D periods can be easily reproduced in our model by choosing parameters in the mixed region where U is meta-stable and D is quasi-stable (*Figure 5A* light orange).

In addition, we found non-zero correlations between consecutive D-U and U-D period durations, a feature that was not observed previously in similar experimental conditions (*Stern et al., 1997*). Reduced statistical power (~30 U-D/D-U pairs were considered by (*Stern et al., 1997*) versus a range of 462–758 pairs in our n = 7 experiments) and different U-D detection methods (intracellular membrane potential thresholding) could be the reasons for this discrepancy.

## Bistability in cortical networks at low firing rates

Bistability in a dynamical system refers to the coexistence of two possible steady states between which the system can alternate (*Angeli et al., 2004*). This principle has been used to interpret UP and DOWN states as two attractors of cortical circuits (*Cossart et al., 2003*; *Shu et al., 2003*) and it seems to underlie higher cognitive functions (*Compte, 2006*; *Durstewitz, 2009*). In particular, multi-stability in recurrent cortical networks has been postulated to underlie the persistent activity observed during the delay period in working memory tasks (*Amit and Brunel, 1997*). Extensive theoretical work has shown that based on the change in curvature of the neuronal $f$ - $I$ curve, that is, from expansive to contractive, recurrent network models generate two types of co-existing attractors: a spontaneous state with arbitrarily low rates (falling in the expansive part of the $f$ - $I$ curve) and a sustained activity attractor where the reverberant activity of a subpopulation of neurons could be maintained at a rate on the contractive part of the $f$ - $I$ curve (*Amit and Brunel, 1997*; *Brunel, 2000a*; *Wang, 2001*). Thus, unless additional mechanisms are included, e.g. synaptic short-term depression and facilitation (*Barbieri and Brunel, 2007*; *Hansel and Mato, 2013*; *Mongillo et al., 2012*) or fined-tuned EI balance (*Renart et al., 2007*), the rate of persistent states is lower-bounded by the rate where the $f$ - $I$ curve changes from convex to concave (~10–20 spikes/s). Moreover, because of this the sustained attractor operates in an unbalanced supra-threshold regime where spike trains tend to be more regular (i.e. lower inter-spike-interval CV, [*Barbieri and Brunel,*

2007; *Hansel and Mato, 2013*; *Renart et al., 2007*]) than those observed in the data (*Compte et al., 2003b*).

UP and DOWN states represent in contrast transitions between very different levels of activity: a quiescent state and a very low rate state. Given that we recorded neurons extracellularly, our estimate of the mean firing rate during UP periods (3.7 spikes/s) is most likely an overestimation. Whole cell intracellular recordings have reported rates in the range 1–2 spikes/s (*Constantinople and Bruno, 2011*), 0.4 spikes/s in Pyramidal L2/3 of the somatosensory cortex of awake mice (*Gentet et al., 2012*), 0.1 spikes/s in Pyramidal L2/3 cells in somatosensory cortex during UP periods in anesthetized rats (*Waters and Helmchen, 2006*), or 0.1–0.3 spikes/s in V1 neurons of awake mice (*Haider et al., 2013*). Juxtacellular recordings have found values near 4–5 spikes/s (*Massi et al., 2012*; *Sakata and Harris, 2009*) whereas Calcium imaging experiments report spontaneous rates < 0.1 spikes/s (*Kerr et al., 2005*). Despite UP rates being so low, rate models have commonly used the change in curvature of the transfer function to generate UP and DOWN dynamics (*Curto et al., 2009*; *Lim and Rinzel, 2010*; *Mattia and Sanchez-Vives, 2012*; *Mochol et al., 2015*). It is also for this reason that most spiking network models generating UP and DOWN transitions exhibit unrealistically high rates during U periods (in the range 10–40 spikes/s) with relatively regular firing (*Bazhenov et al., 2002*; *Compte et al., 2003b*; *Destexhe, 2009*; *Hill and Tononi, 2005*).

An alternative mechanism to generate bistability between UP and DOWN states has been the shunting or divisive effect of inhibitory synaptic conductances, a mechanism that can produce non-monotonic transfer functions and yield bistability between a zero rate state and a state of very low rate (*Kumar et al., 2008*; *Latham et al., 2000*; *Vogels and Abbott, 2005*). Latham and colleagues (*Latham et al., 2000*) addressed the question of how to obtain a state of low firing rate (i.e. <1 spikes/s) in a recurrent EI network and concluded that there were two alternative mechanisms: the most robust was to have a single attractor that relied on the excitatory drive from endogenously active neurons in the network or from external inputs. In fact, excitatory external inputs have been widely used to model low rate tonic spontaneous activity (i.e. no DOWN states) in EI networks of current-based spiking units (*Amit and Brunel, 1997*; *Brunel, 2000b*; *Vogels and Abbott, 2005*). Alternatively, in the absence of endogenous or external drive, a silent attractor appears and a second attractor can emerge at a low rate over a limited range of parameters if inhibition exerts a strong divisive influence on the excitatory transfer function (*Latham et al., 2000*). Based on this, a spiking network of conductance-based point neurons with no external/endogenous activity could alternate between UP (0.2 spikes/s) and DOWN (0 spikes/s) periods via spike frequency adaptation currents. Although the authors did not characterize the statistics of UP and DOWN periods, this network could in principle generate positive correlations between consecutive U and D period durations, Corr(U,D), as long as rate fluctuations caused UP to DOWN transitions for sufficiently different adaptation values (*Lim and Rinzel, 2010*). However, since DOWN to UP transitions were caused by recovery from adaptation, the duration of a D period could not influence the duration of the following U period and their network could not produce correlations between consecutive D-U periods (i.e. Corr(D,U)~0, as in the network shown in *Figure 7—figure supplement 2*). The model moreover lacked bi-modality in the membrane voltage and did not specifically predict a distinct decay of $r_E$ and $r_I$ during UP periods.

Our model proposes a more parsimonious mechanism underlying UP-DOWN bistability: the ubiquitous expansive threshold non-linearity of the transfer function plus the asymmetry in threshold ($\theta_I > \theta_E$) and gain (larger for I than E cells). We used a threshold-linear function for simplicity but other more realistic choices (e.g. threshold-quadratic) produced the same qualitative results. The threshold asymmetry is supported by in vitro patch clamp experiments revealing that firing threshold of inhibitory fast-spiking neurons, measured as the lowest injected current causing spike firing, is higher than that of excitatory regular-spiking neurons (*Cruikshank et al., 2007*; *Schiff and Reyes, 2012*). Inhibition in this model becomes active when external inputs onto E cells during the DOWN state are strong enough to push the system above the separatrix (*Figure 4C*) and ignite the UP state. Once recruited, inhibition is necessary to stabilize the activity because, in its absence, the positive feedback would make the UP state unstable, a condition known as an Inhibition-Stabilized Network (*Ozeki et al., 2009*). In this regime, excitatory currents are supra-threshold but when combined with inhibition result in a net subthreshold input current yielding low-rate irregular firing (*Figure 7—figure supplement 2*).

A direct implication of the specific mechanism of bistability in our model was that intrinsic adaptation of excitatory neurons (*McCormick et al., 1985*) did not cause a noticeable decrease in $r_E$ during the UP periods but instead produced a significant decay in the inhibitory rate $r_I$. We confirmed this prediction in our data (*Figure 6C–D*). Interestingly, the same effect was also observed in ketamine anesthetized animals from both extracellular (*Luczak and Barthó, 2012*) and intracellular recordings resolving synaptic conductances (*Haider et al., 2006*). During DOWN periods, in contrast, the network is not in a balanced state and recovery from adaptation caused a significant increase in the rate of putative excitatory neurons, as predicted by the model. In sum, our results present the first EI network model with linearly summed inputs exhibiting bistability between a quiescent state and a inhibition-stabilized state with arbitrary low rate.

## The role and origin of fluctuations in UP-DOWN switching

Our findings stress the role of input fluctuations inducing transitions between the UP and DOWN network attractors because noise-induced alternations generate periods with large variability as found in the data (*Figure 2A–B*). Adaptation was also necessary to introduce positive serial correlations and to reproduce the observed gamma-like UP-DOWN distributions (compare *Figure 2A–B* with *Figures 5D* and *7H*) because it caused a soft refractory period after each transition decreasing the duration CVs below one (*Figure 2B*). In our rate model fluctuations were simply introduced by a time-varying Gaussian input so that in both DU and UD transitions the noise had the same external origin. In cortical circuits however these two transitions are very different: while in UP-DOWN transitions the fluctuations can originate in the stochasticity of the spiking activity during the UP period, DU transitions depend on either local circuit mechanisms that do not need spiking activity or on external inputs to escape from a quiescent state. Our spiking EI network model could use the stochasticity of the recurrent spiking activity to cause transitions from a low-rate UP state to a quiescent DOWN state but needed synchronous input bumps to cause DOWN to UP transitions (*Figure 7*). Other models have proposed that synaptic noise (e.g. spontaneous miniatures) could cause the transitions from the quiescent state (i.e. DOWN to UP) (*Bazhenov et al., 2002*; *Holcman and Tsodyks, 2006*; *Mejias et al., 2010*). Our analysis shows however that to cause noise-driven transitions from a quiescent state using independent synaptic fluctuations into each cell (1) neurons need to be depolarized unrealistically close to threshold and hence do not display bi-modal voltage distributions, and (2) the magnitude of adaptation must be tuned such that it brings neurons close to threshold allowing the sparse firing to trigger a transition. In this condition moreover, the network does not generate positive correlations between D and consecutive U intervals (*Figure 7—figure supplement 2*). For this reason we used instead synchronous external input *kicks* as the inducers of DOWN-to-UP transitions. These input kicks were also effective driving UP to DOWN transitions but they caused an excess of E and especially I activity at the UP offset (see UP-offset peaks in *Figure 7J*). This feature was not observed in our data but has been observed when triggering UP to DOWN transitions with electrical stimulation (*Shu et al., 2003*). When kicks were suppressed during UP periods, UP-to-DOWN transitions could be triggered by intrinsically generated fluctuations in the spiking activity and the offset peaks in the E and I rates could be largely reduced (not shown). This seems to suggest that the two type of transitions could be triggered by different types of events: DOWN to UP would be triggered by synchronous bumps whereas UP to DOWN by fluctuations in the rates of the two populations. Modeling such a mixed-factors network would require considering more realistic connectivity patterns (e.g. sparse and spatially organized) in order for the network to intrinsically generate more realistic spiking variability in the population (*Amit and Brunel, 1997*; *Vogels and Abbott, 2005*; *Renart et al., 2010*; *Rosenbaum et al., 2017*). For simplicity, we opted for an all-to-all connected network (as opposed to e.g. sparse connectivity) because it was simpler to analyze theoretically and simulate numerically. In particular, the fluctuations of synaptic input in an all-to-all network are set as a fixed parameter, independent of the recurrent activity. This allowed us to find the appropriate network states and determine their stability using standard mean field techniques (*Roxin and Compte, 2016*) and then adjust the magnitude of the fluctuations and kicks to reproduce the transition dynamics using numerical simulations. We leave for future study the extension of these results to more realistic sparse connectivity patterns. In a sparse randomly connected EI network for instance, it would be of interest to study the behavior of this type of bistability as the network size N increases and synaptic couplings are scaled as in a balanced network, that is, $J \sim 1/\sqrt{N}$

(*Renart et al., 2010*; *van Vreeswijk and Sompolinsky, 1998*). In the large N limit, balanced networks linearly transform external inputs into population average output rate (*van Vreeswijk and Sompolinsky, 1998*). This implies that, the larger the network, the more fine tuning of the parameters would be necessary in order to generate this type of bistability.

### Previous evidence supporting membrane voltage synchronous bumps

Evidence for temporally sparse synchronous inputs comes from intracellular membrane potential recordings under some types of anesthesia (pentobarbital or halothane) showing «presynaptic inputs [...] organized into quiescent periods punctuated by brief highly synchronous volleys, or 'bumps'» (*DeWeese and Zador, 2006*). We postulate that these spontaneous bumps (*DeWeese and Zador, 2006*; *Tan et al., 2013*; *Taub et al., 2013*) (1) are caused by synchronous external inputs impinging on the neocortex, possibly from thalamocortical neurons (*Crunelli and Hughes, 2010*), since spontaneous bumps resemble sensory evoked responses (*DeWeese and Zador, 2006*) or from hippocampal Sharp Wave Ripples (*Battaglia et al., 2004*); (2) their timing resembles a Poisson stochastic process rather than a rhythmic input (*Tan et al., 2013*); (3) they lie at the origin of the DOWN-to-UP transitions that we observe. Despite the fact that UP-DOWN-like activity can emerge in cortical slices in vitro (*Cossart et al., 2003*; *Fanselow and Connors, 2010*; *Mann et al., 2009*; *Sanchez-Vives and McCormick, 2000*) the intact brain can generate more complex UP-DOWN patterns than the isolated cortex, with subcortical activity in many areas correlating with transition times (*Battaglia et al., 2004*; *Crunelli et al., 2015*; *Crunelli and Hughes, 2010*; *David et al., 2013*; *Lewis et al., 2015*; *Slézia et al., 2011*; *Ushimaru et al., 2012*). A recent study however reported very large (>20 mV) non-periodic synchronous bumps in cortical in vitro slices (*Graupner and Reyes, 2013*) suggesting that these events could also be generated within local cortical circuits.

These arguments suggest that DOWN to UP transitions are, at least in part, caused by punctuated external synchronous inputs (*Battaglia et al., 2004*; *Johnson et al., 2010*), with slow intrinsic adaptation mechanisms contributing to modulate the probability that these events trigger a transition (*Moreno-Bote et al., 2007*). This complements the view that UP-DOWN dynamics reflect an endogenous oscillation of the neocortex and connects to the role of UP-DOWN states in memory consolidation: because in the active attractor (UP) the *stationary* activity is irregular and asynchronous (*Renart et al., 2010*), the existence of a silent attractor enables synchronous transient dynamics in the form of DOWN to UP transitions. These transients generate precise temporal relations among neurons in a cortical circuit (*Luczak et al., 2007*), which can cause synaptic plasticity underlying learning and memory (*Peyrache et al., 2009*). We speculate that, while the transient dynamics are triggered by external inputs, adaptation, by introducing refractoriness in this process, parses transition events preventing the temporal overlap of information packets (*Luczak et al., 2015*).

## Materials and methods

### Experimental procedures

This study involved analysis of previously published and new data. Previously published data (*Barthó et al., 2004*) was obtained under a protocol approved by the Rutgers University Animal Care and Use Committee. One new data set was performed in accordance with a protocol approved by the Animal Welfare Committee at University of Lethbridge (protocol # 0907). All surgeries were performed under anesthesia, and every effort was made to minimize suffering. Adult, male Sprague-Dawley rats (250–400 g) were anesthetized with urethane (1.5 g/kg) and supplemental doses of 0.15 g/kg were given when necessary after several hours since the initial dose. We also used an initial dose of Ketamine (15–25 mg/kg) before the surgery to induce the anesthetized state quickly. We then performed a craniotomy over the somatosensory cortex, whose position was determined using stereotaxic coordinates. Next 32 or 64 channels silicon microelectrodes (Neuronexus technologies, Ann Arbor MI) were slowly inserted into in deep layers of the cortex (depth 600–1200 μm; lowering speed ~1 mm/hour). Probes had either eight shanks each with eight staggered recording sites per shank (model Buzsaki64-A64), or four shanks with two tetrode configurations in each (model A4 × 2-tet-5mm-150-200-312-A32). Neuronal signals were high-pass filtered (1 Hz) and amplified (1,000X) using a 64-channel amplifier (Sensorium Inc., Charlotte, VT), recorded at 20 kHz sampling rate with 16-bit resolution using a PC-based data acquisition system (United Electronic Industries, Canton,

MA) and custom written software (Matlab Data Acquisition Toolbox, MathWorks) and stored on disk for further analysis.

## Data analysis

Spike sorting was performed using previously described methods (*Harris et al., 2000*). Briefly, units were isolated by a semiautomatic algorithm (http://klustakwik.sourceforge.net) followed by manual clustering procedures (http://klusters.sourceforge.net). We defined the *Population activity* as the merge of the spike trains from all the well isolated units.

## Putative E/I neuronal classification

Isolated units were classified into narrow-spiking (I) and broad-spiking (E) cells based on three features extracted from their mean spike waveforms: spike width, asymmetry and trough-to-peak distance. The two classes were grouped in the space of features by k-means clustering (*Barthó et al., 2004*; *Csicsvari et al., 1998*; *Sirota et al., 2008*).

## Synchronized state assessment

We classified the brain state based on the silence density defined as the fraction of 20 ms bins with zero spikes in the Population activity in 10 s windows (*Mochol et al., 2015*; *Renart et al., 2010*). Epochs with consecutive windows of silence density above 0.4, standard deviation below 0.1 and longer than 5 min, were considered as sustained synchronized brain state and were used for further analysis (synchronized states durations mean ±SD: 494 ± 58 s, n = 7 epochs).

## UP and DOWN transitions detection

UP-DOWN phases have been commonly defined from intracellular recordings by detecting the crossing times of a heuristic threshold set on the membrane potential of individual neurons (*Mukovski et al., 2007*; *Stern et al., 1997*), or from local field potential signals (*Compte et al., 2008*; *Mukovski et al., 2007*) or combined together with the information provided by multi-unit activity (*Haider et al., 2006*; *Hasenstaub et al., 2007*). Defining UP-DOWN phases from single-unit recordings is more challenging because individual neurons fire at low rates discharging very few action potentials on each UP phase (*Constantinople and Bruno, 2011*; *Gentet et al., 2012*; *Waters and Helmchen, 2006*). However, pooling the spiking activity of many neurons into a population spike train reveals the presence of co-fluctuations in the firing activity of the individual neurons and allows accurate detection of UP-DOWN phases (*Luczak et al., 2007*; *Saleem et al., 2010*). We used a discrete-time hidden semi-Markov probabilistic model (HMM) to infer the discrete two-state process that most likely generated the population activity (*Chen et al., 2009*). Thus, the population activity spike count was considered as a single stochastic point process whose rate was modulated by the discrete hidden state and the firing history of the ensemble of neurons recorded. In order to estimate the hidden state at each time, the method used the expectation maximization (EM) algorithm for the estimation of the parameters from the statistical model (*Chen et al., 2009*). Although the discrete-time HMM provides a reasonable state estimate with a rather fast computing speed, the method is restricted to locate the UP and DOWN transition with a time resolution given by the bin size (*T*) for the population activity spike count (10 ms in our case). The initial parameters used for the detection were: Bin-size $T$ = 10 ms, number of history bins J = 2 (sets the length of the memory, i.e. J = 0 is a pure Markov process); history-dependence weight β = 0.01 (i.e. β = 0 for a pure Markov process); transition matrix $P_{DU}$ = $P_{UD}$ = 0.9, $P_{DD}$ = $P_{UU}$ = 0.1; rate during UP periods α = 3, and rate difference during DOWN and UP periods μ = −2 (*Chen et al., 2009*). The algorithm gives an estimate of the state of the network on each bin $T$. Adjacent bins in the same state are then merged to obtain the series of *putative UP (U) and DOWN (D)* periods. The series is defined by the onset $\{t_i^{on}\}_{i=1}^M$ and offset $\{t_i^{off}\}_{i=1}^M$ times of the Us, where M is the total number of Us, that determine the *i*-th UP and DOWN period durations as (see *Figure 1C*):

$$U_i = t_i^{off} - t_i^{on}$$
$$D_i = t_i^{on} - t_{i-1}^{off}$$

(1)

## Statistics of UP and DOWN durations

The mean and the coefficient of variation of $U_i$ were defined as

$$<U_i> = \frac{1}{M}\sum_{i=1}^{M} U_i, \quad CV(U_i) = \frac{\sqrt{Var(U_i)}}{<U_i>} \tag{2}$$

where:

$$Var(U_i) = \left(\frac{1}{M}\sum_{i=1}^{M} U_i^2\right) - <U_i>^2 \tag{3}$$

and equivalently for $<D_i>$ and $CV(D_i)$. We controlled whether variability in $U_i$ was produced by slow drifts by computing $CV_2$ a measure of variability not contaminated by non-stationarities of the data (**Compte et al., 2003a**; **Holt et al., 1996**).

The serial correlation between $U_i$ and $D_{i+k}$, with $k$ setting the lag in the U-D series, e.g. $k = 0$ ($k = 1$) refers to the immediately previous (consecutive) DOWN period, was quantified with the Pearson correlation coefficient defined as:

$$Corr(U_i, D_{i+k}) = \frac{Cov(U_i, D_{i+k})}{\sqrt{Var(U_i)Var(D_i)}} \tag{4}$$

where the covariance was defined as:

$$Cov(U_i, D_{i+k}) = \frac{1}{M-|k|}\sum_{i=1}^{M-k} (U_i - <U_i>)(D_{i+k} - <D_i>) \tag{5}$$

Values of $U_i$ and $D_i$ differing more than 3 standard deviations from the mean were discarded from the correlation analysis (circles in **Figure 2C**). To remove correlations between $U_i$ and $D_i$ produced by slow drifts in the durations we used resampling methods developed to remove slow correlations among spike trains (**Amarasingham et al., 2012**). We generated the *l-th* shuffled series of U periods $\left\{u_i^l\right\}_{i=1}^{M}$ by randomly shuffling the order of the Us in the original series $\{U_i\}_{i=1}^{M}$ within intervals of 30 s. The same was done to define the shuffled series of D periods $\left\{d_i^l\right\}_{i=1}^{M}$. The two shuffled series lack any correlation except that introduced by co-variations in the statistics with a time-scale slower than 30 s. We generated $L = 1000$ independent shuffled series $\left\{u_i^l\right\}_{i=1}^{M}$ and $\left\{d_i^l\right\}_{i=1}^{M}$ with $l=1,2,...L$, computed the covariance $Cov(u_i^l, d_{i+k}^l)$ for each and the averaged over the ensemble $Cov(u_i, d_{i+k}) = <Cov(u_i^l, d_{i+k}^l)>_l$. Finally, the correlation due to co-fluctuations of Us and Ds faster than 30 s was computed by subtracting $Cov(u_i, d_{i+k})$ from $Cov(U_i, D_{i+k})$ in **Equation 5**. Significance of the correlation function $Corr(U_i, D_{i+k})$ was assessed by computing a point-wise confidence interval from a distribution of L correlograms $Corr(u_i^l, d_{i+k}^l)$, for $l = 1...L$ ($L = 10000$), computed from each shuffled series the same way as for the original series (gray dashed bands in **Figure 2C**). To take into account multiple comparisons introduced by the range in lag $k$, we obtained *global* confidence intervals (black dashed bands in **Figure 2C**) by finding the $P$ of the pointwise intervals for which only a fraction of the correlograms $Corr(u_i^l, d_{i+k}^l)$ crosses the interval bands at *any* lag $k= -7...7$ (see **Fujisawa et al., 2008** for details).

Gamma parameter estimates for distributions of U and D durations were computed using the Matlab built-in function gamfit.

## Spike count statistics

We divided the time in bins of $dt = 1$ ms and defined the spike train of the *j*-th neuron as:

$$s_j(t) = \begin{cases} 1 & \textit{if there is a spike} \in (t, t+dt) \\ 0 & \textit{otherwise} \end{cases} \tag{6}$$

The *spike count* of the *j*-th neuron over the time window ($t$-T/2, $t$+T/2) was obtained from

$$n_j(t;T) = \left(K * s_j\right)(t) \tag{7}$$

where $*$ refers to a discrete convolution and $K(t)$ is a square kernel which equals one in (-T/2,T/2) and zero otherwise.

The instantaneous rate of the $j$-th neuron was defined as:

$$r_i(t) = \frac{n_i(t;T)}{T} \tag{8}$$

and therefore the instantaneous population rate was defined as:

$$R(t) = \frac{\sum_{j=1}^{N} n_j(t;T)}{TN} \tag{9}$$

where $N$ is the total number of well isolated and simultaneously recorded neurons. We have dropped the dependence on $T$ from $r_i(t)$ and $R(t)$ to ease the notation. We also defined the instantaneous E-population and I-populations rates, $R^E(t)$ and $R^I(t)$ respectively, as those computed using cells in the E and I subpopulations separately.

## Population firing statistics during Us and Ds

The instantaneous population rate averaged across Us and Ds and aligned at the D to U transition (DU) was defined as:

$$R_{DU}(\tau) = \frac{1}{m(t)} \sum_{i \epsilon \{\tau < U_i\}} R\left(t_i^{on} + \tau\right), \text{ for } \tau > 0 \tag{10}$$

where $\tau$ is the time to the DU transition. Because Us had different durations, for each $\tau > 0$, the sum only included the onset time $t_i^{on}$ if the subsequent period was longer than $\tau < U_i$. By doing this we remove the trivial decay we would observe in $R_{DU}(\tau)$ as $\tau$ increases due to the increasing probability to transition into a consecutive period $D_{i+1}$. For $\tau < 0$, $R_{DU}(\tau)$ reflecting the population averaged rate during the Ds, is obtained as in **Equation 10** but including the times $t_i^{on}$ in the sum if the previous D was longer than $|\tau| < D_{i-1}$. Similarly, the average population rate aligned at the offset $R_{UD}(\tau)$ was defined equivalently by replacing $\{t_i^{on}\}_{i=1}^{M}$ by the series of offset times $\left\{t_i^{off}\right\}_{i=1}^{M}$. We also defined the onset and offset-aligned averaged population rate for excitatory (E) and inhibitory (I) populations, termed $R_{DU}^E(\tau)$ and $R_{UD}^E(\tau)$ for the E case and similarly for the I case. Moreover, the onset and offset-aligned averaged rate of the $i$-th neuron $r_{DU}^i(\tau)$ and $r_{UD}^i(\tau)$ were defined similarly using the individual rate defined in **Equation 8**.

The autocorrelogram of the instantaneous population rate was defined as:

$$AC(\tau) = \frac{\sum_{t=1}^{L-\tau} R(t)R(t+\tau) - <R(t)>_t^2}{(L - |\tau|) \, Var(R(t))}, \quad \text{for } \tau > 0 \tag{11}$$

with the sum in $t$ running over the $L$ time bins in a window of size W. The average $<R(t)>_t$ and variance were performed across time in the same window. To avoid averaging out a rhythmic structure in the instantaneous population rate due to slow drift in the oscillation frequency, we computed $AC(\tau)$ in small windows W = 20 s thus having a more instantaneous estimate of the temporal structure. With the normalization used, the autocorrelograms give $AC(\tau = 0) = 1$ and the values with $\tau > 0$ can be interpreted as the Pearson correlation between the population rate at time $t$ and the population rate at time $t + \tau$ (**Figure 1D**).

## Instantaneous rates at onset and offset intervals

To compare the population rates at the U-onset and U-offset (**Figures 3** and **6**), we computed for each neuron the mean of $r_{DU}^i(\tau)$ over the window $\tau$ = (50,200) s (U-onset) and the mean of $r_{UD}^i(\tau)$ over the window $\tau$ = (−200,−50) s (U-offset). We positioned the windows 50 ms away of the DU and UD transitions in order to preclude the possibility of contamination in the mean rate estimations due to possible misalignments from the U and D period detections. In the averaging we used U and D periods longer than 0.5 s, so that onset and offset windows were always non-overlapping. Equivalent

D-onset and D-offset windows were defined in order to compare individual rates during D periods. To make the distribution of mean rates across the cell population Gaussian, we normalized each of the rates $r_{DU}^i(\tau)$ and $r_{UD}^i(\tau)$ by the overall time-averaged rate of the neuron $r_i = <r_i(t)>_t$ finally obtaining onset and offset-aligned *normalized* averaged rates (e.g. $r_{DU}^i(\tau)/r_i$). Despite this normalization, the distribution of the normalized rates in the D-onset and D-offset was non-Gaussian (most neurons fired no spikes). Thus we used the non-parametric two-sided Wilcoxon signed rank test to compare onset and offset rates (*Figure 3E*). To test the rates changes during U periods in E and I neurons we used a four-way mixed-effects ANOVA with fixed factors onset/offset, E/I and random factors neuron index and animal. We compared the distribution of *normalized* averaged rate difference at the U-onset minus the U -offset (*Figure 3E* right, dark gray histogram) with a distribution obtained from the same neurons but randomly shuffling the onset and offset labels of the spike counts but preserving trial and neuron indices (*Figure 3E* right, light gray bands show 95% C.I. of the mean histograms across 1000 shuffles). This surrogate data set represents the hypothesis in which none of the neurons shows any onset vs offset modulation. The comparison shows that there are significant fractions of neurons showing a rate decrease and increase that compensate to yield no significant difference on the population averaged rate. The same procedure was followed with the normalized rates in the D-onset and D-offset but the limited number of non-zero spike counts limited the analysis yielding inconclusive results (*Figure 3E* left).

## Computational rate model

We built a model describing the rate dynamics of an excitatory ($r_E$) and inhibitory population ($r_I$) recurrently connected that received external inputs (*Wilson and Cowan, 1972*). In addition, the excitatory population had an additive negative feedback term $a(t)$, representing the firing adaptation experienced by excitatory cells (*McCormick et al., 1985*). The model dynamics were given by:

$$\tau_E \frac{dr_E}{dt} = -r_E(t) + \varphi_E(J_{EE}r_E(t) - J_{EI}r_I(t) - a(t) + \sigma\xi_E(t)) \tag{12}$$

$$\tau_I \frac{dr_I}{dt} = -r_I(t) + \varphi_I(J_{IE}r_E(t) - J_{II}r_I(t) + \sigma\xi_I(t)) \tag{13}$$

$$\tau_a \frac{da}{dt} = -a(t) + \beta r_E(t) \tag{14}$$

The time constants of the rates were $\tau_E$= 10 ms and $\tau_I$= 2 ms, while the adaptation time constant was $\tau_a$= 500 ms. The synaptic couplings $J_{XY} > 0$ (with $X,Y$ = E, I), describing the strength of the connections from $Y$ to $X$, were $J_{EE}$= 5, $J_{EI}$= 1, $J_{IE}$= 10, $J_{II}$= 0.5 s. Because we are modeling low rates, the adaptation grows linearly with $r_E$ with strength $\beta$= 0.5 s. The fluctuating part of the external inputs $\sigma\xi_X(t)$ was modeled as two independent Ornstein–Uhlenbeck processes with zero mean, standard deviation $\sigma$= 3.5 and time constant 1 ms for both E and I populations. Because population averaged firing rates during spontaneous activity fell in the range 0–10 spikes/s, we modeled the transfer functions $\varphi_X$ as threshold-linear functions:

$$\varphi_X(x) = g_X [x - \theta_X]_+ \quad , X = \{E,I\} \tag{15}$$

where the square brackets denote $[z]_+ = z$ if $z > 0$ and zero otherwise, the gains were $g_E$ = 1 Hz and $g_I$= 4 Hz and the *effective* thresholds $\theta_E$ and $\theta_I$ represented the difference between the activation threshold minus the mean external current into each population. We took $\theta_I$= 25 a.u. and explored varying $\theta_E$ over a range of positive and negative values (*Figure 5A–B*). The choice of thresholds $\theta_E < \theta_I$ and gains $g_E < g_I$ reflecting the asymmetry in the *f-I* curve of regular spiking neurons (E) and fast spiking interneurons (I) (*Cruikshank et al., 2007*; *Nowak et al., 2003*; *Schiff and Reyes, 2012*), facilitated that the model operated in a bistable regime (see below).

Input-output transfer functions are typically described as sigmoidal-shaped functions (*Haider and McCormick, 2009*), capturing the nonlinearities due to spike threshold and firing saturation effects. Since we are interested in modeling spontaneous activity where average population rates are low, we constrained the transfer functions to exhibit only an expanding non-linearity reflecting the threshold and thus avoid other effects that can only occur at higher rates (the contracting non-linearity

tends to occur for rates >30 spikes/s (*Anderson et al., 2000*; *Houweling et al., 2010*; *Nowak et al., 2003*; *Priebe and Ferster, 2008*). In particular, we modeled $\varphi_X$ as piecewise linear (*Schiff and Reyes, 2012*; *Stafstrom et al., 1984*) but the same qualitative bistable regime can be obtained by choosing for instance a threshold-quadratic function. The model equations (*Equations 12-14*) were numerically integrated using a fourth-order Runge-Kutta method with integration time step dt = 0.2 ms. U and D periods in the model were detected by threshold-based method, finding the crossing of the variable $r_E$ with the boundary 1 Hz, where periods shorter than minimum period duration of 50 ms were merged with neighboring periods (small changes in threshold and period durations did not affect qualitatively the results). The computational rate model was implemented in Matlab (MathWorks) using C ++ MEX, and the source code is available at ModelDB (https://senselab.med.yale.edu/ModelDB, *Jercog, 2017*).

## Fixed points and stability

We start by characterizing the dynamics of the system in the absence of noise. Assuming that the rates evolve much faster than the adaptation, that is, $\tau_E, \tau_I \ll \tau_a$, one can partition the dynamics of the full system into (1) the dynamics of the rates assuming adaptation is constant, (2) the slow evolution of adaptation assuming the rates are constantly at equilibrium. Thus, the equations of the *nullclines* of the 2D rate dynamics at fixed *a*, can be obtained from the 2D system given by *Equations 12-13*. The nullclines of this reduced 2D system are obtained by setting its left hand side to zero:

$$r_E = g_E [\, J_{EE}\, r_E - J_{EI} r_I - a - \theta_E\,]_+ \tag{16}$$

$$r_I = g_I [\, J_{IE}\, r_E - J_{II}\, r_I - \theta_I\,]_+ \tag{17}$$

The intersection of the nullclines define the fixed points $(r_E^*(a), r_I^*(a))$ of the 2D system to which the rates evolve. Once there adaptation varies slowly assuming that the rates are maintained at $(r_E^*(a), r_I^*(a))$ until it reaches the equilibrium at $a = \beta\, r_E^*(a)$.

The network has a fixed point in $(r_E, r_I, a) = (0,0,0)$ if and only if $\theta_E \geq 0$ and $\theta_I \geq 0$, that is, when the mean external inputs are lower than the activation thresholds. The stability of this point, corresponding to the DOWN state, further requires $\theta_E > 0$, thus preventing the activation of the network due to small (infinitesimal) fluctuations in $r_E$. To find an UP state fixed point with non-zero rates we substitute in *Equations 16-17* the value of adaptation at equilibrium $a = \beta r_E$, assume the arguments of $[]_+$ are larger than zero and solve for $(r_E, r_I)$, obtaining:

$$r_E = \frac{1}{|M|}\, (\, J_{EI}\, \theta_I - J_{II}'\, \theta_E) \tag{18}$$

$$r_I = \frac{1}{|M|}\, ((J_{EE}' - \beta)\, \theta_I - J_{IE}\, \theta_E) \tag{19}$$

where $|M| = J_{EI}\, J_{IE} - (J_{EE}' - \beta)(J_{II}')$, $J_{EE}' = J_{EE} - \frac{1}{g_E}$ and $J_{II}' = J_{II} + \frac{1}{g_I}$.

The conditions for this UP state solution to exist are derived from imposing that the right hand side of *Equations 18-19* is positive. The stability of this solution (*Equation 21* below) implies that the determinant $|M|$ is positive and that if $r_I$ is positive, then $r_E$ is also positive. Thus, provided the stability (*Equations 21-22*), the only condition for the solution to exist is that the right hand side of *Equation 19* is positive:

$$\theta_E < \frac{(J_{EE}' - \beta)}{J_{IE}}\, \theta_I \tag{20}$$

Given the separation of time scales described above, this fixed point is stable if the eigenvalues of the matrix of coefficients of *Equations 16 and 17* without the term *a* (that we assume is constant) have all negative real part. Because the coefficients matrix is 2 × 2, this is equivalent to impose that the determinant of the matrix has a positive determinant and a negative trace. These conditions yield the following inequalities, respectively:

$$J_{II}{}' \, J_{EE}{}' < J_{EI} \, J_{IE} \tag{21}$$

$$\tau_I(g_E J_{EE} + 1) < \tau_E(g_I J_{II} + 1) \tag{22}$$

*Equation 21* is equivalent to the condition that the I-nullclines of the 2D reduced system has a larger slope than the E-nullcline. From the U existence condition in *Equation 20* and D stability condition, it can also be derived that $J_{EE}{}' > 0$, implying that at fixed inhibition, the E-subnetwork would be unstable (i.e. slope of the E-nullcline is positive). In sum, the conditions for the existence of two stable U and D states imply that the U state would be unstable in the absence of feedback inhibition but the strength of feedback inhibition is sufficient to stabilize it. These are precisely the conditions that define an Inhibitory Stabilized Network state (*Ozeki et al., 2009*).

## Phase plane analysis

In this section we determine the different operational regimes of the network in the $(\theta_E, \beta)$-plane (*Figure 5A*). In the absence of noise, given that $\theta_I \geq 0$, a stable D state exists in the semi-plane (*Figure 5A*, purple and red regions):

$$\theta_E > 0 \tag{23}$$

Provided that our choice of synaptic couplings $J_{XY}$ and time constants hold the stability conditions (*Equations 21-22*), the U state is stable in the semi-plane given by *Equation 20* (*Figure 5A*, orange and red regions):

$$\beta < -\frac{J_{IE}}{\theta_I} \, \theta_E + J_{EE}{}' \tag{24}$$

In the intersection of these two semi-planes both D and U are stable (bistable region, *Figure 5A* red). In contrast, in the complementary region to the two semi-planes, neither U nor D are stable (*Figure 5A* white region). There, a rhythmic concatenation of relatively long U and D periods is observed where the network stays transiently in each state until adaptation triggers a transition (see e.g. *Figure 4E*). Because of the separation of time-scales, we refer to this stability to the rate dynamics but not to the adaptation dynamics as *quasi-stable* states.

The addition of noise makes that some of the stable solutions now become meta-stable, meaning that the network can switch to a different state by the action of the noise (i.e. the external fluctuations in our model). This is the case of the bistable region (*Figure 5A* red) where fluctuations generate stochastic transitions between the two metastable U and D states (*Figure 4D*). In the region of D stability $\theta_E > 0$, we find a new subregion with noise-driven transitions from a metastable D state to a *quasi-stable U state*, and back to D by the action of adaptation (*Figure 5A* light purple). This subregion is delimited by the condition that U is not stable (i.e. *Equation 24* does not hold) but *just* because of the existence of adaptation. This can be written by saying that *Equation 24* holds if $\beta = 0$:

$$\theta_E < \frac{J_{EE}{}'}{J_{IE}} \, \theta_I \tag{25}$$

Equivalently, within the region of U stability, noise creates a new subregion with noise-driven transitions from a metastable U state to a *quasi-stable D state*, and back to U by the recovery from adaptation (*Figure 5A* light orange). This subregion is given by the condition that there is a negative effective threshold $\theta_E < 0$ (i.e. caused by a supra-threshold mean external drive) but the adaptation $a^U$ recruited in the U state is sufficient to counterbalance it: $a^U + \theta_E > 0$. This makes the D transiently stable until adaptation decays back to zero. Substituting $a^U = \beta r_E^U$ (*Equation 14*) and $r_E^U$ by the equilibrium rate at the U state given by *Equation 18*, the limit of this subregion can be expressed as (*Figure 5A*, light orange region):

$$\beta > \frac{(J_{EE}{}' \, J_{II}{}' - J_{IE} \, J_{EI})}{J_{EI} \, \theta_I} \, \theta_E \tag{26}$$

## Spiking network simulations

We used a network model of leaky integrate-and-fire neurons (*Ricciardi, 1977*), with $N_E$=4000 excitatory and $N_I$=1000 inhibitory neurons 'all-to-all' connected. The membrane potential of a neuron $i$ from population $E$ and $I$ obeys

$$\tau_E \frac{dV_i^E}{dt} = -\left(V_i^E - V_L\right) + I_{rec}^E(t) + I_{ext,i}^E(t) - I_{a,i}(t) \tag{27}$$

$$\tau_I \frac{dV_i^I}{dt} = -\left(V_i^I - V_L\right) + I_{rec}^I(t) + I_{ext,i}^I(t) \tag{28}$$

Whenever the membrane voltage of a $X = \{E, I\}$ neuron exceeds the threshold $\theta_X$ at time $t$, a spike is emitted and the membrane voltage is reset to $V_r^X$, that is, whenever $V_i^X(t^-) \geq \theta_X$ then $V_i^X(t^+) = V_r^X$. We used $V_r^E$ = -51 mV, $V_r^I$ = −49.9 mV and a leak potential of $V_L$ = −57.4 mV. The thresholds were $\theta_E$ = −45 mV and $\theta_I$ = −43.9 mV, and we used no spike refractory time. The membrane time constants were $\tau_E$ = 20 ms and $\tau_I$ = 10 ms. The external input current $I_{ext,i}^X(t) = \sigma\sqrt{\tau_X}\eta_i(t) + p_i I_{kicks}(t)$ is composed of: (1) a Gaussian white noise term with std. dev. $\sigma$ = 2.5 mV which is independent from neuron to neuron, that is, $<\eta_i(t)\,\eta_j(t-t')> = \delta_{ij}\delta(t-t')$ and was necessary to generate uncorrelated firing across neurons given the all-to-all connectivity. (2) A separate source of randomly occurring input pulses, also called 'kicks', impinging coherently on 10% of both E and I neurons in the network ($p_i$ is a binary random variable with probability p=0.1):

$$I_{kicks}(t) = K \sum_k \left(1 - e^{-\frac{t-t^k}{\tau_k}}\right)\chi_\Delta\left(t - t^k\right) \tag{29}$$

with $K$ being the pulses amplitude (220 mV during D, 110 mV during U for E cells; 88 mV during D, 44 mV during U for I cells), $\tau_k$ the rise time (0.5 ms) of the pulses and $\chi_\Delta(t)$ the step function defined as 1 in the interval $(0, \Delta)$ and zero otherwise. We used duration $\Delta$ = 2 ms and amplitude $K$ causing a depolarization of 16.1 mV (6.44 mV) during D (U) periods in E-kicked neurons, and 12.4 mV (4.9 mV) during D (U) periods in I-kicked neurons. These kicks were necessary to generate synchronous bumps in the membrane potential that would yield transitions to the UP state during DOWN periods in the absence of background activity (see Discussion). The amplitude of these events was constant for the sake of simplicity.

The recurrent input term consisted of inhibitory and excitatory synaptic currents, that is, $I_{rec}^X(t) = I_{rec}^{XE}(t) + I_{rec}^{XI}(t)$, where $I_{rec}^{XY} = J_{XY}s_Y(t)$ and $J_{XY}$ is the synaptic strength from neurons in population $Y$ to neurons in population $X$. The synaptic variable $s_X$ obeyed the following differential equation

$$\tau_d^X \frac{ds_X}{dt} = -s_X + u_X \tag{30}$$

$$\tau_r^X \frac{du_X}{dt} = -u_X + \underline{\tau} \sum_k \sum_{j=1}^{N_X} \delta\left(t - t_j^k - d_j^X\right) \tag{31}$$

where the summation is over all spikes emitted by all neurons in population $X$ (all-to-all connectivity) and the factor $\underline{\tau} = 1$ ms ensures that the area under the unitary synaptic event is constant regardless of the rise and decay time-constants. The synaptic couplings were $J_{EE}$ = 1.4, $J_{EI}$ = −0.35, $J_{IE}$ = 5 and $J_{II}$ = −1 mV and the rise $(\tau_r^X)$ and decay $(\tau_d^X)$ times of inhibitory synapses were both 1 ms, while those of excitation were 8 and 23 ms, respectively. These synaptic kinetic constants were chosen in order to reduce the magnitude of the fast oscillations during UPs. The delays $d_j^X$ were the same for all the postsynaptic synapses belonging to the same neuron and uniformly distributed between 0 and 1 ms (0 and 0.5 ms) across E (I) neurons.

In addition, the excitatory neurons displayed an after hyperpolarization (AHP) current $I_a$ that follows

$$\tau_a \frac{dI_{a,i}}{dt} = -I_{a,i} + \beta \sum_k \delta\left(t - t_i^k\right) \tag{32}$$

with a slow adaptation time constant $\tau_a$ = 500 ms and adaptation strength $\beta$ = 300 mVms.

The population averaged AHP current is defined as

$$I_a(t) = \sum_{i=1}^{N_E} \frac{I_{a,i}(t)}{N_E} \tag{33}$$

The E and I nullclines of the population averaged rates $r_E$ and $r_I$, respectively, are obtained from the equilibrium firing rate ($r_{0X}$), which is given by the self consistent mean-field equation (*Ricciardi, 1977*; *Amit and Brunel, 1997*)

$$r_{0X} = \left[ 2\tau_X \int_{(V_r - \mu_{0X})/\sigma_{0X}}^{(\theta - \mu_{0X})/\sigma_{0X}} du\, e^{u^2} \int_{-\infty}^{u} dv\, e^{-v^2} \right]^{-1} \tag{34}$$

where the average input current to a neuron in population $X$ is given by

$$\mu_{0E} = V_L + N_E J_{EE} r_{0E} \underline{\tau} + N_I J_{EI} r_{0I} \underline{\tau} + I_a \tag{35}$$

$$\mu_{0I} = V_L + N_E J_{IE} r_{0E} \underline{\tau} + N_I J_{II} r_{0I} \underline{\tau} \tag{36}$$

and the standard deviation of the current is given by the external white noise std. dev. $\sigma$. For this analysis the AHP current $I_a$ was assumed to be constant at the averaged values observed either at the UP onset and offset (see dark and light red E-nullclines in *Figure 7A*, respectively). The bifurcation diagrams (*Figure 7B* and *Figure 7—figure supplement 2B*) were obtained by solving the stationary states of the network from the Fokker Planck equation describing the population dynamics and determining their stability using linear perturbation analysis (*Brunel and Hakim, 1999*; *Richardson, 2007*; *Roxin and Compte, 2016*).

## Analysis of spiking network simulations

The spiking network model was implemented in C++, and the code is available at ModelDB (https://senselab.med.yale.edu/ModelDB, *Jercog, 2017*). Model equations were numerically integrated using second-order Runge Kutta, where integration step was defined as dt = 0.05 ms and the total simulation length was 5000 s (50 simulations of 100 s each, where data from first D and U detected periods for each simulation was discarded to eliminate possible initial transient effects). For the analysis of U and D statistics and the rate dynamics in the spiking network simulations we used methods analogous to those applied on the experimental data. In addition, U and D period detection was obtained by applying the HMM (*Chen et al., 2009*) on 100 randomly selected neurons, using the same parameters as for the experimental data. Onset and offset aligned population rates $r_E$ and $r_I$ (*Figure 7J*) were computed using randomly sampled 90 E and 10 I neurons, respectively, with a minimum U/D period duration of 0.8 s.

In order to study the statistics of UP and DOWN dynamics in the spiking network, where transitions are caused by independent noise among cells (*Figure 7—figure supplement 2*), external kicks were not included and E cells were depolarized by 5.6 mV to keep their voltage right below their spike threshold. In addition, to keep the UP firing rate stabilized at low values, I cells were depolarized by 6 mV and synaptic decay time constant for excitatory synapses were set to $\tau_r^E$ = 2 ms and $\tau_d^E$ = 3 ms. In addition, adaptation strength was set to $\beta$ = 200 mVms, and HMM detection parameters chosen as $\alpha$ = 2 and $\mu$ = −1. The rest of the parameters are the same as those used in main *Figure 7*.

## Acknowledgements

We thank Zhe Chen for sharing the HMM based UP-DOWN detection related scripts, Narani van Laarhoven for sharing codes, Kenneth Harris, Gyuri Buzsáki and the members of the Compte and de

la Rocha labs for helpful discussions. This work was supported by the AGAUR of the Generalitat de Catalunya (Ref: SGR14-1265), the Spanish Ministry of Economy and Competitiveness together with the European Regional Development Fund (grants BFU2009-09537, BFU2012-34838 to AC, RYC-2011–08755 to AR, SAF2010-15730, SAF2013-46717-R and RYC-2009–04829 to JR), the EU (Marie Curie grants PIRG07-GA-2010–268382 to JR and BIOTRACK contract PCOFUND-GA-2008–229673 to AR), Hungarian Brain Research Program Grant (KTIA_NAP_13-2-2014-0016 to PB). Part of the work was carried out at the Esther Koplowitz Centre, Barcelona.

## Additional information

### Funding

| Funder | Grant reference number | Author |
|---|---|---|
| Ministerio de Economía y Competitividad | Spanish Ministry of Economy and Competitiveness together with the European Regional Development Fund: RYC-2011-08755 | Alex Roxin |
| European Regional Development Fund | Spanish Ministry of Economy and Competitiveness together with the European Regional Development Fund: RYC-2011-08755 | Alex Roxin |
| EU Biotrack contract | PCOFUND-GA-2008-229673 | Alex Roxin |
| Hungarian Brain Research Program Grant | KTIA_NAP_13-2-2014-0016 | Peter Barthó |
| Ministerio de Economía y Competitividad | Spanish Ministry of Economy and Competitiveness together with the European Regional Development Fund: BFU2009-09537 | Albert Compte |
| European Regional Development Fund | Spanish Ministry of Economy and Competitiveness together with the European Regional Development Fund: BFU2009-09537 | Albert Compte |
| Ministerio de Economía y Competitividad | Spanish Ministry of Economy and Competitiveness together with the European Regional Development Fund: BFU2012-34838 | Albert Compte |
| European Regional Development Fund | Spanish Ministry of Economy and Competitiveness together with the European Regional Development Fund: BFU2012-34838 | Albert Compte |
| Agència de Gestió d'Ajuts Universitaris i de Recerca | SGR14-1265 | Albert Compte |
| Ministerio de Economía y Competitividad | Spanish Ministry of Economy and Competitiveness together with the European Regional Development Fund: SAF2010-15730 | Jaime de la Rocha |

| European Regional Development Fund | Spanish Ministry of Economy and Competitiveness together with the European Regional Development Fund: SAF2010-15730 | Jaime de la Rocha |
|---|---|---|
| Ministerio de Economía y Competitividad | Spanish Ministry of Economy and Competitiveness together with the European Regional Development Fund: RYC-2009-04829 | Jaime de la Rocha |
| European Regional Development Fund | Spanish Ministry of Economy and Competitiveness together with the European Regional Development Fund: RYC-2009-04829 | Jaime de la Rocha |
| Ministerio de Economía y Competitividad | Spanish Ministry of Economy and Competitiveness together with the European Regional Development Fund: SAF2013-46717-R | Jaime de la Rocha |
| European Regional Development Fund | Spanish Ministry of Economy and Competitiveness together with the European Regional Development Fund: SAF2013-46717-R | Jaime de la Rocha |
| European Commission | EU Marie Curie grants: PIRG07-GA-2010-268382 | Jaime de la Rocha |
| Ministerio de Economía y Competitividad | SAF2015-70324R | Jaime de la Rocha |

The funders had no role in study design, data collection and interpretation, or the decision to submit the work for publication.

## Author contributions

Daniel Jercog, Software, Investigation, Methodology, Writing—original draft, Writing—review and editing, Data Analysis; Alex Roxin, Software, Methodology, Writing—review and editing; Peter Barthó, Artur Luczak, Data acquisition; Albert Compte, Conceptualization, Investigation, Writing—review and editing, Data analysis; Jaime de la Rocha, Conceptualization, Investigation, Writing—original draft, Writing—review and editing, Data Analysis

## Author ORCIDs

Daniel Jercog http://orcid.org/0000-0003-3849-9196
Jaime de la Rocha http://orcid.org/0000-0002-3314-9384

## Ethics

Animal experimentation: This study involved analysis of previously published and new data. Previously published data (Bartho et al, J Neurophys. 2004, 92(1)) was obtained under a protocol approved by the Rutgers University Animal Care and Use Committee. One new data set was acquired in accordance with a protocol approved by the Animal Welfare Committee at University of Lethbridge (protocol # 0907). All surgeries were performed under anesthesia, and every effort was made to minimize suffering.

## Decision letter and Author response

Decision letter https://doi.org/10.7554/eLife.22425.017
Author response https://doi.org/10.7554/eLife.22425.018

## Additional files

**Supplementary files**
• Transparent reporting form.
DOI: https://doi.org/10.7554/eLife.22425.016

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
