## [Decision Letter]

Thank you for submitting your article "UP-DOWN cortical dynamics reflect state transitions in a bistable balanced network" for consideration by *eLife*. Your article has been favorably evaluated by Eve Marder (Senior Editor) and three reviewers, one of whom is a member of our Board of Reviewing Editors. The following individuals involved in review of your submission have agreed to reveal their identity: Benjamin Lindner (Reviewer #2); Brent Doiron (Reviewer #3).

The reviewers have discussed the reviews with one another and the Reviewing Editor has drafted this decision to help you prepare a revised submission.

Summary:

This paper is devoted to further our understanding of up and down states in neural firing as observed in different brain states. Both novel experimental data and a simple yet novel computational rate model are presented allowing the authors to draw conclusions about the mechanism of up/down states and transitions between them.

The authors compare two possible mechanisms for the emergence of up/down states: (a) cellular adaptation that destabilizes each of the states on the time scale of seconds or (b) a true bistable situation, in which transitions between metastable states are triggered by external or internal fluctuations. Mechanism (a) seems to be at odds with their finding of little variation in the population rate when conditioned on the transition time. Also the high variability of the durations is more in line with a fluctuation-driven transition.

They then develop a simple computational model for the firing rates of excitatory and inhibitory subpopulations, which explain their experimental findings They predict a pronounced decline in the rate of the inhibitory neurons during the UP state and, by re-analyzing their data, can confirm this nontrivial prediction.

Overall, all reviewers thought this was a clearly written and solid study which, as it is an important contribution to basic cortical models, would interest a wider community, as the mechanism seems quite general.

Essential revisions:

One question is whether the conclusion would carry over directly to spiking networks. Did the authors search for the experimentally observed features also in a network of spiking neurons (one remark in the Discussion seems to indicate this)? Some of them have certainly great experience with these kinds of models and so it is not clear whether the features seen are difficult to achieve in a spiking-neuron model, or whether they are only typical for rate models. If the mechanisms illustrated are so basic that it would not be difficult to see them in a network of spiking neuron models, the authors could add one more figure illustrating this for the most salient features seen in their experimental data.

Figure 6 seems to be making the case that the up-down transitions seen in data are due to the fact that the cortex lives in the noise-induced transition state. However, in the model bifurcation diagram (Figure 5, panels A and B), we see that even the region where the up to down transition is adaptation induced rather than noise induced, it is still possible to get reasonable CVs for residence times in the up state (approx. 0.5). One would naively expect that as the transitions become more adaptation (rather than fluctuation) induced, the CV should shrink dramatically. However, that could be parameter dependent.

However since the CV can be high in the U to D adaptation regime it then begs the question: What would the analogue of Figure 6 panel A look like in that regime rather than the fully noise induced regime? Basically, Figure 6 currently provides some positive evidence for suggesting the U to D transitions should be noise induced, but no negative evidence that they shouldn't be adaptation induced.

In the Discussion the authors point out that Latham et al. developed a network model with adapting conductance-based point neuron models that displays transitions between a (practically silent) down state and a low-firing (0.2 Hz) up state, i.e. it achieves one goal of the simple rate model though in a more biophysically realistic setting. The authors point out that their own model is more 'parsimonious' and can reflect some of their own experimental findings. However, it was not clear to us whether the Latham-model would be able to show the same effects or not. The authors should add a statement on this point.

---

## [Author Response]

*Essential revisions:*

*One question is whether the conclusion would carry over directly to spiking networks. Did the authors search for the experimentally observed features also in a network of spiking neurons (one remark in the Discussion seems to indicate this)? Some of them have certainly great experience with these kinds of models and so it is not clear whether the features seen are difficult to achieve in a spiking-neuron model, or whether they are only typical for rate models. If the mechanisms illustrated are so basic that it would not be difficult to see them in a network of spiking neuron models, the authors could add one more figure illustrating this for the most salient features seen in their experimental data.*

This has been a major addition to the manuscript. We have included a new section in the Results with three new figures:

Figure 7 showing the main results of the spiking network composed of LIF neurons;

Figure 7—figure supplement 1, with more details about how transitions are obtained in the network;

Figure 7—figure supplement 2, with a network working on a different transitions regime (similar to the network used in Latham et al. 2000).

In brief, we were able to reproduce the main results of the rate model regarding the statistics of UP and DOWN periods (Figure 7: rates during UP and DOWN periods, mean durations  and <D>, CV(U) and CV(D) and Corr(U,E) and Corr(D,U). Moreover, the model reproduced the more accentuated decay of the *r_I_* compared to the *r_E_* along UP periods (Figure 7), despite the fact that adaptation mechanisms were only present in E-cells. To match the statistics of UP-DOWN dynamics in experimental data, external large punctuated inputs causing large synchronous bumps had to be injected in the network to cause DOWN to UP transitions (see Results). There is actually ample evidence of bump-like events in cortical intracellular recordings both in vivo (see e.g. deWeese and Zador 2006) and in vitro (Graupner and Reyes, 2014), as we discuss in the Discussion section.

In the absence of these kicks, DOWN to UP transition could only be generated by depolarizing neurons close to threshold during the DOWN period and tuning adaptation such that the recovery from adaptation would set the system in a saddle-node bifurcation where it can easily transition from the DOWN to the UP (see Figure 7—figure supplement 2). In this regime of network operation, intracellular voltages are not bi-modal as typically described in intracellular studies of UP and DOWN dynamics, and the simulation does not show the observed pattern of correlations between successive periods of U and D intervals.

These results are described in the last two paragraphs of the Results section.

Some aspects of this spiking network, however, do not comply with the data. The most visible is the population surge of inhibitory activity at the UP offset (Figure 7), a feature previously observed in slices when DOWN to UP transitions are externally driven (Shu et al. 2003) but not in our data (Figure 6). Moreover, the all-to-all connectivity is also an aspect of the spiking model which could be further improved with a sparsely connected network to match more aspects of the data. We consider however, that the focus of this paper was on the statistics of UP and DOWN duration and on the increase or decrease of population averaged rates during these periods, and thus leave other aspects such as the structure of the spiking variability during UP periods for future work. These limitations of the model are discussed in the section “The role and origin of fluctuations in UP-DOWN switching”in the Discussion.

*Figure 6 seems to be making the case that the up-down transitions seen in data are due to the fact that the cortex lives in the noise-induced transition state. However, in the model bifurcation diagram (Figure 5, panels A and B), we see that even the region where the up to down transition is adaptation induced rather than noise induced, it is still possible to get reasonable CVs for residence times in the up state (approx. 0.5). One would naively expect that as the transitions become more adaptation (rather than fluctuation) induced, the CV should shrink dramatically. However, that could be parameter dependent.*

*However since the CV can be high in the U to D adaptation regime it then begs the question: What would the analogue of Figure 6 panel A look like in that regime rather than the fully noise induced regime? Basically, Figure 6 currently provides some positive evidence for suggesting the U to D transitions should be noise induced, but no negative evidence that they shouldn't be adaptation induced.*

Here, the referees raise two different points. On the one hand, the relationship between the different firing rate decay rates of excitatory and inhibitory neurons (Figure 6) and the regime of operation of the network (adaptation or fluctuation regimes). On the other hand, the distinction between these two regimes. We will first address this last point.

The limit-cycle (adaptation regime) and bi-stable (fluctuation-driven regime) regimes are only qualitatively different in a noiseless system (Figure 5). Noise makes the transition between these two phases become smooth and progressive, making the difference only quantitative (see new Figure 5—figure supplement 1). What it is clear however, is that independently of whether one starts it in the limit cycle phase or in the bi-stable phase, the system requires large amounts of input fluctuations to trigger the transitions in an irregular fashion (CVs grow generally with σ in Figure 5—figure supplement 1). We think that this point is valid and important: independently of whether the system is in one regime or in the other fluctuations are important to drive these dynamics. In addition, both in our rate model and in the spiking network, we could only match the statistics of UP and DOWN periods, when putting the system in the bi-stable regime. This is because as adaptation grows stronger, CVs of UP and DOWN periods drop, and correlations between consecutive interval durations disappear (Figure 5). This depends of course on parameters, especially on the magnitude of the noise σ, so that increasing the noise can recover the irregular statistics. Quantitatively, however, we could not achieve some of the statistics by increasing σ from the adaptation regime. Particularly constraining was the CV(D) which was above 0.5, and the fact that strong cross-correlation of successive intervals had a non-monotonic dependence with noise, and decayed for large σ. We have included a new supplementary figure (Figure 5—figure supplement 1) showing that in the rate model these values could only be attained well into the fluctuation driven regime. In addition, there is also a maximum σ beyond which the transfer function becomes effectively linear (i.e. no threshold non-linearity) and the bi-modality of the network’s rates is lost, thus eliminating state-alternating dynamics altogether. As a result, one cannot obtain CVs of arbitrarily high magnitude in the limit-cycle regime by simply cranking up σ, and in our hands one could not attain the CV and CC values observed experimentally other than well into the fluctuation-driven regime (Figure 5). In sum, even if the distinction between fluctuation and adaptation driven regimes becomes blurry in the presence of noise, our investigations with two different network models converge in identifying the fluctuation-driven regime as the one that can most easily replicate the irregular dynamics observed in the experiment.

Regarding the comment of the referees about how would Figure 6 look like in the adaptation driven regime. This is shown in Figure 4. The asymmetric decay of rate E and rate I is also there because this feature is not providing evidence about whether transitions are noise-driven or adaptation-driven but it is providing evidence on the right-shift of the I-nullcline and its large slope, two features than in our model readily follow from the asymmetry in thresholds ϴ*_I_*>ϴ*_E_* and gains *g_I_* > *g_E_*.

We have added one sentence to specify this in the Results subsection “Dynamics of E and I populations during UP and DOWN states: model and data”:

“Note that this feature of the model is not dependent on its specific regime of operation, as it would similarly apply in an adaptation-driven regime (Figure 4).”

*In the Discussion the authors point out that Latham et al. developed a network model with adapting conductance-based point neuron models that displays transitions between a (practically silent) down state and a low-firing (0.2 Hz) up state, i.e. it achieves one goal of the simple rate model though in a more biophysically realistic setting. The authors point out that their own model is more 'parsimonious' and can reflect some of their own experimental findings. However, it was not clear to us whether the Latham-model would be able to show the same effects or not. The authors should add a statement on this point.*

The Latham et al. (2000) model does not describe irregular UP and DOWN dynamics but purely adaptation driven. This is because they are interested in “isolated networks” with no external inputs. An isolated network, they conclude, requires “endogenously active neurons” which in our framework means neurons that, when recovered by adaptation, are sufficiently close to threshold to fire independently of their synaptic activity. This is very similar to the spiking network that we have included in Figure 7—figure supplement 2. This network shows serial correlations between consecutive U and D periods, but it does not show correlations between consecutive D and U period durations (see Figure 7—figure supplement 2). This is because correlation between D and the consecutive U requires of a stochastic mechanism to escape the DOWN period that cannot be provided by only adaptation recovery. We think the Latham network would produce the same correlation pattern as the one shown in our new Figure 7—figure supplement 2.

Regarding the different decay of E and I rates during UP periods, the model by Latham et al. (2000) does not seem to have a clear prediction. In some of the toy nullclines drawn in the paper (E.g. Figure 4) there is a slight shift in the I-nullcline which could create some asymmetry in the decays Δ*r_E_* and Δ*rI*. However, in other examples (e.g. Figure 4) the asymmetry is no longer there. Thus, the model by Latham et al. (2000) does not predict univocally a more accentuated firing rate adaptation in I-cells than E-cells, and in fact their figures show prominent firing rate adaptation in E-cells (their Figure 6, 9F, 11).

In the Discussion we have included two explicit sentences about whether the Latham model could reproduce our finding:

“Alternatively, in the absence of endogenous or external drive, a silent attractor appears and a second attractor can emerge at a low rate over a limited range of parameters if inhibition exerts a strong divisive influence on the excitatory transfer function (Latham et al., 2000). Based on this, a spiking network of conductance-based point neurons with no external/endogenous activity could alternate between UP (0.2 spikes/s) and DOWN (0 spikes/s) periods via spike frequency adaptation currents.”

Although the authors did not characterize the statistics of UP and DOWN periods, this network could in principle generate positive correlations between consecutive U and D period durations, Corr(U,D), as long as rate fluctuations caused UP to DOWN transitions for sufficiently different adaptation values (Lim and Rinzel, 2010). However, since DOWN to UP transitions were caused by recovery from adaptation, the duration of a D period could not influence the duration of the following U period and their network could not produce correlations between consecutive D-U periods (i.e. Corr(D,U)~0, as in the network shown in Figure 7—figure supplement 2). The model moreover lacked bi-modality in the membrane voltage and did not specifically predict a distinct decay of *r_E_* and *r__ _I_* during UP periods.__